# Structural analyses define the molecular basis of clusterin chaperone function

Patricia Yuste-Checa [1,2] ✉, Alonso I. Carvajal[1], Chenchen Mi[1], Sarah Paatz[1], F. Ulrich Hartl [1,2,3] ✉ & Andreas Bracher [1] ✉

Clusterin (apolipoprotein J), a conserved glycoprotein abundant in blood and cerebrospinal fluid, functions as a molecular chaperone and apolipoprotein. Dysregulation of clusterin is linked to late-onset Alzheimer disease. Despite its prominent role in extracellular proteostasis, the mechanism of clusterin function remained unclear. Here, we present crystal structures of human clusterin, revealing a discontinuous three-domain architecture. Structure-based mutational analysis demonstrated that two disordered, hydrophobic peptide tails enable diverse activities. Resembling the substrate-binding regions of small heat-shock proteins, these sequences mediate clusterin's chaperone function in suppressing amyloid-β, tau and α-synuclein aggregation. In conjunction with conserved surface areas, the tail segments also participate in clusterin binding to cell surface receptors and cellular uptake. While contributing to lipoprotein formation, the hydrophobic tails remain accessible for chaperone function in the lipoprotein complex. The remarkable versatility of these sequences allows clusterin to function alone or bound to lipids in maintaining the solubility of aberrant extracellular proteins and facilitating their clearance by endocytosis and lysosomal degradation.

Extracellular molecular chaperones are implicated in numerous pathologies including Alzheimer disease (AD) and other neurodegenerative disorders associated with aggregate deposition[1]. The conserved vertebrate secretory glycoprotein clusterin (Clu, also known as apolipoprotein J (ApoJ)) functions as an abundant extracellular chaperone in mammalian blood plasma, seminal fluid and cerebrospinal fluid[2–5]. Initially described as a factor mediating cell clustering in vitro[6,7], Clu was also shown to have an immune-regulatory function by inhibiting complement-mediated cell lysis[8,9] and fractionating with high-density lipoprotein-like particles in plasma and the central nervous system[10–14]. Characterization of Clu as a molecular chaperone is based on its activity in vitro to inhibit the aggregation of non-native proteins[15] and neurodegenerative disease proteins amyloid-β (Aβ)[16–20], tau[21,22], α-synuclein[22,23] and prion protein[24,25]. Consistent with a role as a chaperone, Clu is upregulated in AD and colocalizes with Aβ deposits

in the brain, possibly reflecting a role in the clearance of misfolded proteins by facilitating receptor-mediated endocytosis and lysosomal degradation[26–28]. Indeed, certain *CLU* gene alleles rank among the most significant genetic risk factors for late-onset AD[29–31].

Clu is synthesized as a ~50-kDa precursor protein with a secretory signal peptide that is removed upon translocation into the endoplasmic reticulum (ER), where six *N*-glycans are attached. The protein chain is cleaved into the β-chain and α-chain (residues 23–227 and 228–449, respectively) by furin-like proteases in the Golgi (Fig. 1). The chains remain covalently linked by five conserved disulfide bonds[32]. To provide a basis for detailed mechanistic studies, we determined the crystal structure of human Clu and explored its function by rational mutagenesis. We identified two hydrophobic flexible tails as being critical for binding non-native protein, receptor-mediated endocytosis and, in conjunction with an amphipathic helix, formation of lipoprotein

[1]Department of Cellular Biochemistry, Max Planck Institute of Biochemistry, Martinsried, Germany. [2]Aligning Science Across Parkinson's (ASAP) Collaborative Research Network, Chevy Chase, MD, USA. [3]Munich Cluster for Systems Neurology (SyNergy), Munich, Germany. ✉e-mail: yuste@biochem.mpg.de; uhartl@biochem.mpg.de; bracher@biochem.mpg.de

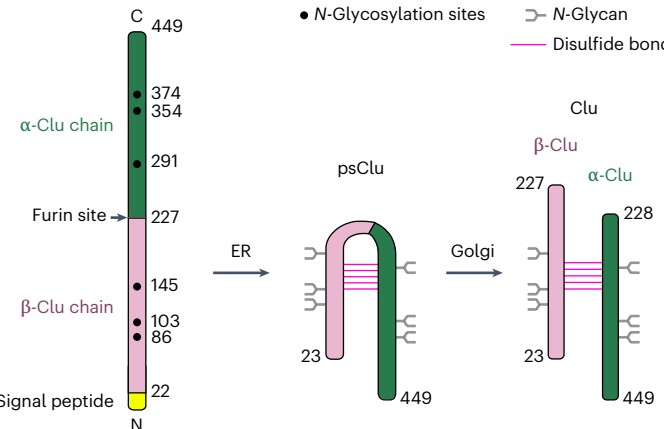

**Fig. 1 | Biogenesis of secreted Clu.** The Clu precursor is thought to be cotranslationally inserted into the ER lumen followed by signal peptide (yellow) removal. In the ER, six *N*-glycans are attached and five antiparallel disulfide bonds are formed[32]. During passage through the Golgi network, the *N*-glycans are processed and a furin-like protease cleaves Clu into the β-chain and α-chain (pink and dark green). *N*-glycans and disulfide bonds are indicated by forks (gray) and horizontal lines (magenta), respectively. *N*-glycosylation sites, the furin site and chain termini are indicated.

particles. Interestingly, Clu maintains functionality as a chaperone when bound to lipid.

## Results

### Crystal structure of Clu

Human Clu was overexpressed in HEK293E cells and purified from the cell culture supernatant. The protein was monomeric at pH 5 (apparent size ~90 kDa) but additionally formed dimers of ~220 kDa at pH 6.5–8.5 (Extended Data Fig. 1a) without measurably affecting Clu secondary-structure content and stability (Extended Data Fig. 1b,c). Purified Clu exhibited *N*-glycan heterogeneity and almost complete processing into the α-chain and β-chain migrating at ~40 kDa on SDS–PAGE (Extended Data Fig. 1d). For crystallization, we designed a single-chain construct (Clu-Δ(214–238)) comprising a deletion of a predicted unstructured sequence including the furin cleavage site. Upon expression in the presence of the α-mannosidase I inhibitor kifunensine[33] to generate uniform oligomannose *N*-glycans (Extended Data Fig. 1d,e), purified Clu-Δ(214–238) crystallized at acidic pH in two distinct lattice types, crystal forms I and II, diffracting to 2.8-Å and 3.5-Å resolution, respectively (Table 1). The structures were solved by molecular replacement using the AlphaFold2 model of Clu as a search template (https://www.alphafold.ebi.ac.uk/entry/P10909). Crystal form I contained one copy of Clu-Δ(214–238) while crystal form II contained two independent copies (Fig. 2a and Extended Data Fig. 2a). The three conformations in the crystal lattices differed slightly in their domain orientations, with root-mean-square deviation (r.m.s.d.) values of 1.5–2.0 Å (Cα positions) (Extended Data Fig. 2b). The AlphaFold2 model showed larger deviations, mainly because of domain reorientations, with r.m.s.d. values of 4.7–5.6 Å (Cα positions) (Extended Data Fig. 2c).

The Clu structure is composed of three domains, a coiled-coil (CC) domain, a small helical domain containing all disulfide bridges (hereafter disulfide domain, DD) and a C-terminal α/β roll-like (AB) domain (Fig. 2a,b). The ~85-Å-long CC bundle is composed of helices α2 and α6 from the β-chain and α10 from the α-chain in wild-type (WT) Clu. The CC interactions of these helices form the majority of interchain contacts (Fig. 2a,c). Helix α2 runs antiparallel to helices α6 and α10. The shorter seven-turn helix α7 aligns with the bundle, widening the distance between helices α6 and α10. The five interchain disulfides connect helix α3 and its flanking linkers with the short helices α8, η3 and α9, which zigzag along helix α3 (Fig. 2a,b). Helices α4 and α5

## Table 1 | Crystallographic data collection and refinement statistics

| | Crystal form I | Crystal form II[a] |
|---|---|---|
| **Data collection** | | |
| Space group | $P2_1$ | $C2$ |
| Cell dimensions | | |
| $a, b, c$ (Å) | 65.74, 43.81, 102.80 | 194.44, 46.44, 155.17 |
| $α, β, γ$ (°) | 90, 107.29, 90 | 90, 127.20, 90 |
| Resolution (Å) | 2.8 (2.95–2.8) [b] | 3.5 (3.83–3.5) [b] |
| $R_{merge}$ | 0.127 (1.360) | 0.200 (1.413) |
| $I/σI$ | 5.9 (0.7) | 6.1 (1.4) |
| Completeness (%) | 98.2 (96.1) | 99.8 (99.6) |
| Redundancy | 3.7 (3.6) | 7.2 (5.7) |
| **Refinement** | | |
| Resolution (Å) | 2.8 | 3.5 |
| No. reflections | 13814 | 14331 |
| $R_{work}/R_{free}$ | 0.2322/0.2742 | 0.2352/0.2836 |
| No. atoms | | |
| Protein | 3,007 | 5,726 |
| Ligands and ions | 123 | 238 |
| Water | – | |
| B factors | | |
| Protein | 79.42 | 135.42 |
| Ligands and ions | 119.48 | 182.16 |
| Water | – | – |
| R.m.s.d. | | |
| Bond lengths (Å) | 0.003 | 0.003 |
| Bond angles (°) | 0.522 | 0.505 |

[a]Data from two crystals were merged for processing of crystal form II. [b]Values in parentheses are for highest-resolution shell.

buttress these helices in the α-chain. While the DD seems fairly rigid, its linkers to helices α2 and α10 deviate considerably between crystal forms, indicating structural plasticity (Extended Data Fig. 2b). The AB domain consists of a three-stranded antiparallel β-sheet, which wraps around helix α12 (Fig. 2a). The sheet and helix are flanked by helix α1 (Fig. 2a,b). The AB domain, thus, also contributes to the interchain contacts in WT Clu.

The *N*-glycans in Clu-Δ(214–238) point toward solvent channels in the crystal lattices and are partially ordered (gray spheres, Fig. 2a). These hydrophilic elements function presumably in stabilizing the protein in solution. Three of the *N*-glycans are located in the DD, while the other three are arranged along the CC bundle, together making up ~30% of protein mass (Fig. 2c and Extended Data Fig. 1d). Importantly, the model of the WT structure contains extended unstructured regions, including disordered tails at the C terminus of the β-chain (residues 199–227) and at the N terminus of the α-chain (residues 228–244) (Fig. 2b,c). These tails are generated by furin cleavage in the unstructured sequence of WT Clu that is deleted in Clu-Δ(214–238). The α7–α8 loop (residues 261–280) is also disordered (Fig. 2a,b). Thus, helix α7 (residues 244–257) is flanked by flexible regions in WT Clu.

To rule out structural artifacts resulting from the deletion in the crystallized construct, we analyzed both Clu-Δ(214–238) and WT Clu by hydrogen–deuterium exchange combined with mass spectrometry (H/DX–MS) at pH 5, where the proteins are monomeric (Extended Data Fig. 1a,e and 3). WT Clu generally displayed slow deuterium incorporation in regions that formed secondary structure in Clu-Δ(214–238)

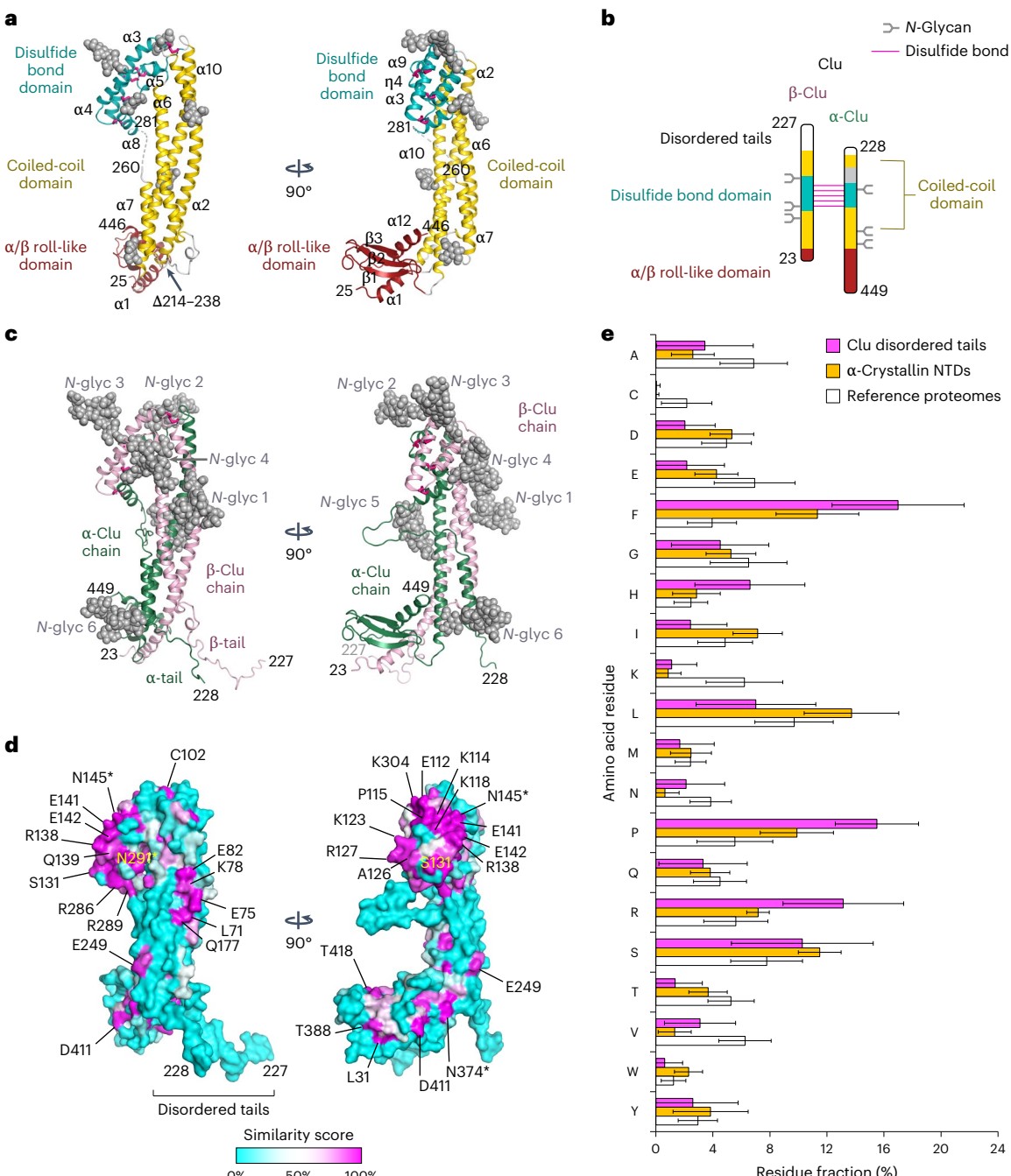

**Fig. 2 | Structure of Clu. a**, Crystal structure of Clu-Δ(214–238). The asymmetric unit of crystal form I is shown in orthogonal views. The CC, DD and AB domains are indicated in gold, teal and dark red, respectively. The dashed line indicates a disordered loop, residues 261–280. Residue numbering refers to the human Clu precursor sequence. Secondary-structure elements are shown in ribbon representation. Disulfide bonds are shown as purple sticks and partially ordered N-glycan structures in space-filling mode are shown in gray. The first and last structured residues, deletion site and secondary-structure elements of Clu are indicated. **b**, Domain architecture of Clu. Simplified scheme of the discontinuous domain structure of Clu in the style of Fig. 1. Domains are colored as in **a**. The disordered tails and the disordered loop are colored white and gray, respectively. **c**, Structural model for WT Clu. The Clu β-chain and α-chain are colored pink and dark green, respectively. The unstructured residues 204–227, 228–238 and 261–280 were modeled in arbitrary conformations.

The full N-glycans (N-glyc 1–6, gray spheres) were modeled as oligomannose trees. **d**, Sequence conservation mapped onto the molecular surface of Clu. The same views as in **c** are shown. A cyan–white–magenta color gradient indicates increasing surface conservation based on the similarity score from the sequence alignment shown in Extended Data Fig. 4. Highly conserved residues are indicated. Asterisks indicate N-glycosylation sites. **e**, Sequence bias in the unstructured tails of Clu. A histogram of the amino acid composition of residues 204–238 in 242 representative Clu sequences is shown in magenta. For comparison, the average compositions of the NTDs in α-crystallin A and B homologs (orange; 662 and 444 sequences, respectively) and the average composition of selected vertebrate reference proteomes (white; human, opossum, platypus, chicken, clawed frog, coelacanth, zebrafish and ghost shark) are shown. Error bars designate the s.d.

(Extended Data Fig. 3a–c). Residues 24–33 at the N terminus, residues 191–195, 220–226 and 234–240 in the predicted flexible tails and residues 270–276 in the disordered α7–α8 loop showed rapid deuterium incorporation in both WT Clu and Clu-Δ(214–238) (Extended Data Fig. 3b,c), consistent with the structural model of WT Clu derived from the Clu-Δ(214–238) crystal structures (Fig. 2c).

## Regions of functional relevance

Clu homologs share ~15% sequence identity and ~40% similarity (Extended Data Fig. 4). Areas of high surface conservation are restricted to the DD (Fig. 2d). We noticed that the disordered tails at the C terminus of the β-chain (β-tail) and the N terminus of the α-chain (α-tail) (Fig. 2c,d), while lacking sequence conservation, display a clear compositional bias toward the hydrophobic amino acid phenylalanine (Fig. 2d,e). Other residues overrepresented in this region are histidine, arginine, proline and serine, with proline and serine known to be generally enriched in loop regions[34]. Lysine and cysteine residues were less frequent than in reference proteomes. Interestingly, a similar sequence bias is found in the flexible N-terminal domains (NTDs) of small heat-shock protein (sHsp) chaperones, such as α-crystallin from vertebrates, which are known to interact with non-native client proteins[35–37] (orange bars in Fig. 2e). Hydrophobic clefts reminiscent of substrate protein-binding sites in molecular chaperones such as Hsp60 and Hsp70 are absent in the Clu crystal structure. However, we detected two areas of hydrophobic contacts between Clu molecules in the crystal lattice (Extended Data Fig. 5a,b); V26, M34, Q37 and V389 in the AB domain flank a hydrophobic crystal contact, while I46, V50, V369, A373 and T376 in the CC domain contact the artificial peptide linkage in Clu-Δ(214–238). Moreover, the narrow cleft between the DD and the CC domain might provide a binding site for an extended peptide after slight opening.

In addition to Clu-Δ(214–238), henceforth referred to as tail mutant TL1, we generated a series of Clu mutant proteins targeting the flexible tail sequences (mutants TL2–TL4), conserved surface residues in DD1–DD3, the amphipathic helix α7 (CC1) and putative substrate-binding regions in CC2–CC4 and AB1 and AB2 (Fig. 3a). In the cases of DD1 and DD2, we introduced multiple substitutions to maximize functional effects while maintaining overall surface properties such as net charge and hydrophobicity. Circular dichroism (CD) spectra indicated helical structure and thermal stability similar to WT Clu for most mutants (Extended Data Fig. 5c,d).

## Structural basis of chaperone function

To identify regions in Clu that mediate the interaction with non-native client proteins, we used rhodanese (Rho) as a model substrate. Rho (33 kDa) rapidly forms amorphous aggregates upon dilution from denaturant, as detected by a turbidity assay (Fig. 3b). WT Clu at 1:1 and 3:1 molar ratios to denatured Rho (D-Rho) efficiently suppressed aggregation at pH 7.2 (Fig. 3b) and formed soluble complexes with D-Rho of ~700 kDa, apparently representing D-Rho oligomers bound to multiple molecules of Clu (Extended Data Fig. 6a). This 'holdase' activity was markedly diminished at pH 5.2, where Clu dimers are absent (Extended Data Figs. 1a and 6b), suggesting that dimer formation is functionally relevant by enhancing the avidity of Clu for client protein aggregates.

Strikingly, deletion of residues 214–238 in mutant TL1 (Fig. 3a), comprising most of the flexible tail sequences, almost completely abolished the ability of Clu to prevent D-Rho aggregation (Fig. 3b,c). To exclude that this loss of chaperone function was because of the single-chain nature of TL1, we generated the furin-site mutant TL4, resulting in an otherwise WT single-chain version of Clu (Fig. 3a). TL4 showed almost normal holdase activity (Fig. 3c), indicating that the loss of aggregation prevention in TL1 is because of the lack of the tail residues. Similarly, Clu mutants in which aromatic and hydrophobic residues in the β-tail and α-tail were substituted by serine (residues 204–227 and 228–238 in TL2 and TL3, respectively, and combined in

TL2 + 3; Fig. 3a) showed reduced aggregation prevention to an extent correlating with the number of substitutions (Fig. 3c). The only other mutant with a similar aggregation prevention defect for D-Rho was DD1 (Fig. 3c), containing multiple substitutions in a conserved surface patch of the DD. Mutants CC3 and CC4 with altered surface clefts at the CC domain had intermediate defects (Fig. 3c). The hydrophobic pocket targeted by mutants AB1 and AB2 does not seem to contribute significantly to interactions with D-Rho (Fig. 3c). Thus, the hydrophobic residues in the flexible tails of Clu are critical for chaperone function and cooperate with the DD and adjacent regions in the interaction of Clu with small, soluble aggregates of a non-native protein.

To extend the analysis of Clu chaperone function to amyloid aggregates and a medically relevant client, we next monitored the effect of Clu mutants on the nucleation-dependent formation of Aβ(1–42) fibrils. Clu has been shown to be effective at substoichiometric concentrations in delaying Aβ amyloid formation[19,38]. Indeed, WT Clu at a ratio to Aβ of 1:300 increased the half-time of amyloid formation[39] from ~1.3 h to ~4 h (Fig. 3d). In contrast, the TL1 mutant was almost completely inactive in delaying Aβ aggregation (Fig. 3d,e). Mutants TL2, TL3 and TL2 + 3 retained partial activity (Fig. 3e). The deletion mutant CC1, which lacks helix α7 and two short sequence motifs proposed to interfere with Aβ aggregation[40], was fully active (Fig. 3e) and so were mutants CC3, CC4 and DD3 (Fig. 3e). However, the single-chain mutant TL4 and mutants CC2 and DD2 were partially defective in delaying Aβ formation, in contrast to their largely unimpaired activity in preventing D-Rho aggregation (Fig. 3c,e). Thus, the flexible hydrophobic tails gain full activity in Aβ aggregation prevention only after furin cleavage and function cooperatively with regions outside the tails (Fig. 3e). Surprisingly, mutant DD1 was hyperactive in the Aβ aggregation assay (Fig. 3e), extending the aggregation half-time from ~4 h for WT Clu to over 12 h (Extended Data Fig. 6c). The mutations in DD1 essentially scramble highly conserved residues in a surface patch, thereby apparently enhancing the contribution of this domain to Aβ aggregation prevention, although reducing the interaction with the heterologous substrate D-Rho (Fig. 3c). Of note, the DD1 site did not act autonomously, as combining DD1 with TL1 completely abolished Aβ aggregation inhibition (Extended Data Fig. 6d). Mutants TL1 and TL2 + 3 were also defective in slowing amyloid formation of α-synuclein and the repeat domain of tau (TauRD)[22,23] (Extended Data Fig. 6e–h).

To assess the effect of the flexible tail regions of Clu in isolation, we transplanted the β-tail and α-tail onto the chain termini of GFP (GFP–TL). In addition, we generated GFP fusions carrying either only the β-tail or α-tail (GFP–βTL and GFP–αTL, respectively; Fig. 4a). GFP itself has no known chaperone function and its N and C termini are similar in distance to Clu residues 204 and 238, from which the β-tail and α-tail emanate. While GFP alone had no effect, GFP–TL was nearly as effective as WT Clu in delaying amyloid formation (Fig. 4b,c). In contrast, the single-tail constructs GFP–βTL and GFP–αTL were essentially without effect, indicating that the tails in GFP–TL must interact synergistically with Aβ aggregates. Interestingly, GFP–TL was ineffective in suppressing D-Rho aggregation (Extended Data Fig. 6i), presumably because of the absence of the complementary DD region and the glycans. The latter confer high Clu solubility, which may be necessary to maintain oligomeric Rho aggregates in solution.

Taken together, our data establish a critical synergistic role of the disordered hydrophobic tails for the interactions of Clu with aggregation-prone client proteins and peptides. While partially effective in aggregation prevention on their own, these regions functionally cooperate with surfaces in the DD. Apparently, mutation of the DD can optimize Clu for prevention of Aβ formation.

## Role of flexible tails in Clu uptake

Clu is thought to facilitate cellular uptake of bound substrate proteins by receptor-mediated endocytosis using various surface receptors[26–28,41]. We monitored Clu uptake with fluorescently labeled Clu (Clu–A488),

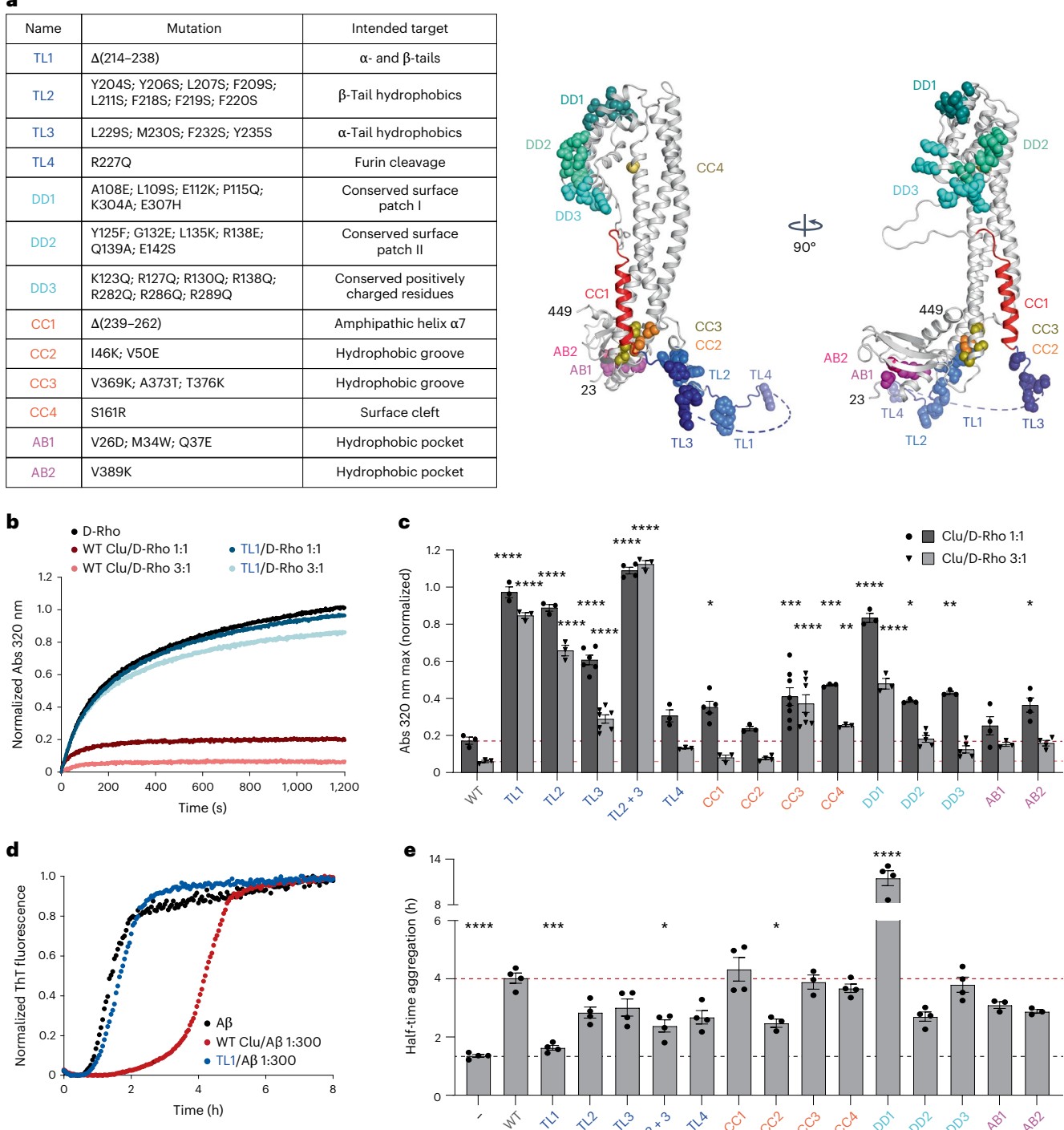

**Fig. 3 | Structural dissection of Clu chaperone activity. a,** Overview of structure-guided mutations in Clu. Left, mutants are named and colored according to their location: TL, blue; DD, cyan; CC, orange; AB, purple. Right, the mutants are mapped onto the Clu structural model. Substituted residues are shown in space-filling mode; deleted regions are marked by a colored backbone ribbon. **b,c,** Effect of Clu mutants on aggregation of D-Rho. D-Rho was diluted into buffer containing no protein, WT Clu or mutants at Clu/D-Rho molar ratios of 1:1 or 3:1. D-Rho aggregation was monitored by turbidity. **b,** Representative normalized absorbance traces of D-Rho alone (black), with additional WT Clu (1:1, dark red; 3:1, light red) and with TL1 (1:1, dark blue; 3:1, light blue). **c,** Quantification of the absorbance plateau determined by curve fitting in the presence of respective Clu mutants, normalized to D-Rho alone (red and light-red dashed lines indicate WT Clu/D-Rho molar ratios at 1:1 and 3:1, respectively). Data represent averages ± s.e.m. (*n* = 3 biological replicates, except for: TL2 + 3, AB1

and AB2, *n* = 4; CC1, *n* = 5; TL3, *n* = 6; CC3, *n* = 8). *$P$ < 0.05, **$P$ < 0.01, ***$P$ < 0.001 and ****$P$ < 0.0001 according to one-way ANOVA with Dunnett's post hoc test comparing Clu mutant/D-Rho versus WT Clu/D-Rho at each concentration ratio. **d,e,** Effect of Clu mutants on Aβ(1–42) amyloid formation. Aβ(M1–42) amyloid formation was monitored by ThT fluorescence in absence or presence of WT Clu or mutants (molar ratio Clu/Aβ: 1:300). **d,** Representative normalized fluorescence curves in absence (black) or presence of WT Clu (red) or TL1 (blue). **e,** Relative delay of Aβ aggregation by WT Clu and mutants determined by the half-time of the aggregation plateau (gray and red dashed lines indicate values for Aβ alone and with WT Clu, respectively). Data represent averages ± s.e.m. (*n* = 4 biological replicates, except for: CC2, CC3, AB1 and AB2, *n* = 3). *$P$ < 0.05, ***$P$ < 0.001 and ****$P$ < 0.0001 according to one-way ANOVA with Dunnett's post hoc test comparing the samples to WT Clu/Aβ.

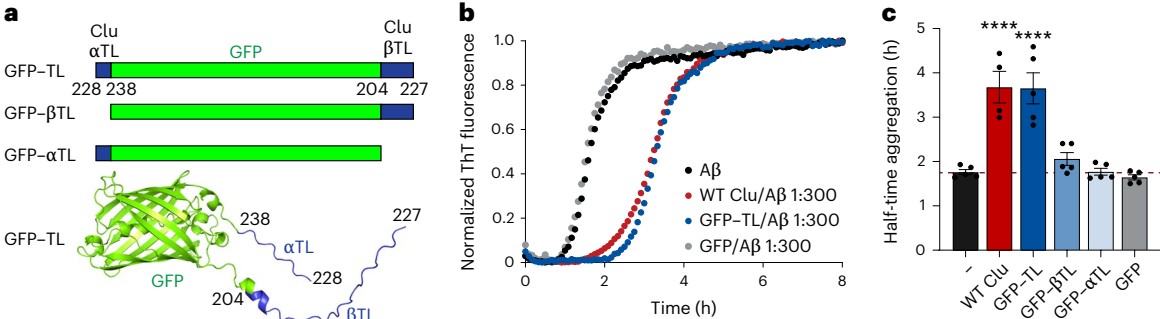

**Fig. 4 | Effect of isolated Clu hydrophobic tails on Aβ formation. a**, Domain structures of GFP−TL, GFP−βTL and GFP−αTL. A structural model of GFP−TL is shown underneath. **b**, Aβ amyloid formation was monitored as in Fig. 3d in the absence (black) or presence of WT Clu (red), GFP (gray) or GFP−TL (blue) (molar ratio Clu/Aβ or GFP/Aβ: 1:300). Representative normalized ThT fluorescence curves are shown. **c**, Half times of aggregation of WT Clu and GFP constructs compared to Aβ alone (red dashed line). Data represent averages ± s.e.m. ($n = 5$ biological replicates, except for: WT Clu, $n = 4$). ****$P < 0.0001$ according to one-way ANOVA with Dunnett's post hoc test comparing Clu or GFP/Aβ to Aβ (WT Clu, $P = 8.45 \times 10^{-6}$; GFP−TL, $P = 4.23 \times 10^{-6}$).

in the absence or presence of the client Aβ aggregates, into induced pluripotent stem cell (iPS cell)-derived neurons (iNeurons) (Extended Data Fig. 7a). Uptake of Clu−A488 was competed by unlabeled Clu (Extended Data Fig. 7b). In the presence of Aβ aggregates, WT Clu uptake was stimulated in a concentration-dependent manner (Fig. 5a), similar to previous observations with denatured firefly luciferase as the substrate[26]. We next monitored the uptake of Clu mutants. Interestingly, the TL1, TL2 and TL2 + 3 mutants exhibited strong uptake defects in the absence and presence of Aβ aggregates (~15−27% and ~29−37% residual uptake relative to WT Clu or WT Clu/Aβ, respectively), with milder defects of TL3 (Fig. 5b). Thus, the same structural elements mediating client protein binding are also critical in cellular uptake. Notably, most other mutants also showed mild to intermediate uptake defects including a previously described variant corresponding to mutant DD3 in the present study[26], with a ~55% reduction in uptake in the absence of substrate (Fig. 5b). These findings may reflect the complexity of Clu binding to a variety of cellular receptors[26–28,41]. Thus, multiple regions in Clu, including the flexible tail sequences, are involved in cellular uptake, possibly by mediating interactions with different surface receptors. Apart from a general stimulation of uptake, the presence of Aβ aggregates had little differential effect on the uptake efficiency of Clu mutants. Mutants with strong uptake defects in the absence of Aβ tended to experience the largest uptake stimulation in the presence of Aβ, consistent with an increase in avidity for receptor binding because of the presence of multiple Clu molecules in complex with client protein oligomers.

To corroborate the critical role of the flexible tails for cellular Clu uptake, we studied the uptake of the fusion proteins GFP−TL, GFP−βTL and GFP−αTL (Fig. 4a) into iNeurons. Compared to GFP alone, the GFP−TL construct bearing both tails showed strong stimulation of internalization (Fig. 5c), confirming the Clu mutant data (Fig. 5b). The GFP fusions with single Clu tails, GFP−βTL and GFP−αTL, exhibited weak uptake stimulation not significantly different from GFP alone (Fig. 5c), suggesting cooperation of both tails in cellular uptake.

We next analyzed the interaction of Clu mutants with very-low-density lipoprotein receptor (VLDLR), an LDL-type receptor implicated in Clu uptake[41,42] (Extended Data Fig. 7c). We expressed and purified the VLDLR ectodomain (VLDLR-ed), which includes the ligand-binding domain and a regulatory motif[43] (Extended Data Fig. 7c). Clu interacted directly with VLDLR-ed as shown by VLDLR-ed immunoprecipitation (Fig. 5d). We used ELISA with immobilized VLDLR-ed to estimate affinities for WT Clu and mutants. WT Clu exhibited single-site binding characteristics with a dissociation constant ($K_D$) of ~80 nM (Fig. 5e). For comparison, we tested the binding of the receptor-associated protein (RAP) (Extended Data Fig. 7c), an ER-resident chaperone known to compete with ligand binding to LDL receptors[41,44]. RAP showed higher

affinity ($K_D$ of ~1.2 nM)[45,46] (Extended Data Fig. 7d) and competed with Clu for VLDLR-ed binding (Extended Data Fig. 7e), suggesting overlapping interaction sites in the VLDLR ligand-binding domain (Extended Data Fig. 7c).

Binding to VLDLR-ed was disrupted in the Clu TL1, TL2 and TL2 + 3 mutants (Fig. 5e and Extended Data Fig. 7f), confirming that the tail sequences directly participate in receptor binding. Binding was more affected for the β-tail mutant TL2 than the α-tail mutant TL3, consistent with the β-tail making a greater contribution to binding (Extended Data Fig. 7f). Interestingly, the previously described uptake mutant DD3 (ref. 26) and the partially overlapping DD2 mutant also exhibited strongly reduced binding affinity for VLDLR-ed, similar in magnitude to the effect of mutant TL3, suggesting the existence of a secondary receptor interaction site in the DD (Extended Data Fig. 7g). All other tested Clu mutants, with the exception of the deletion mutant CC1, showed VLDLR-ed affinities close to WT Clu (Extended Data Fig. 7f–h).

To independently analyze the interactions of the Clu tail sequences with VLDLR-ed, we again used the GFP fusion proteins (Fig. 4a). GFP−TL and GFP−βTL but not GFP−αTL and GFP exhibited clear interactions with resin-bound VLDLR-ed (Extended Data Fig. 7i), supporting the direct participation of the tail sequences in receptor binding.

In summary, the hydrophobic disordered tails of Clu function in receptor-mediated endocytosis. For binding to VLDLR, these sequences cooperate with accessory regions in the DD distinct from the sites contributing to chaperone activity. Thus, there is an overlap in binding regions for chaperone activity and cellular uptake. Nevertheless, the interaction with client protein enhances rather than impedes Clu uptake, suggesting that binding of multiple Clu molecules (dimers or higher oligomers) to small aggregates allows sufficient Clu tails to remain available for association with receptors. Additionally, client binding may increase the affinity of other regions of Clu, such as the DD, for cell surface receptors.

## Formation of Clu−phospholipid particles

Lipoprotein particles containing Clu are present in human blood plasma and cerebrospinal fluid[10,14]. To investigate the structural basis of Clu−lipid binding, we formed complexes of Clu with the neutral phospholipid 1,2-dimyristoyl-*sn*-glycero-3-phosphocholine (DMPC)[28,47]. Analysis by native PAGE revealed the formation of a high-molecular-weight Clu−lipid complex above a critical molar ratio of Clu/DMPC of ~1:500 (Fig. 6a), similar to the behavior of apolipoproteins ApoA1 and ApoE (Extended Data Fig. 8a,b). In contrast, the molecular chaperone Hsc70 did not form complexes with DMPC (Extended Data Fig. 8c). The Clu−DMPC complexes fractionated at ~1–3 MDa in size-exclusion chromatography (SEC) and negative-stain electron microscopy revealed oval disc-shaped particles of ~19 and ~26 nm in diameter (Fig. 6b and

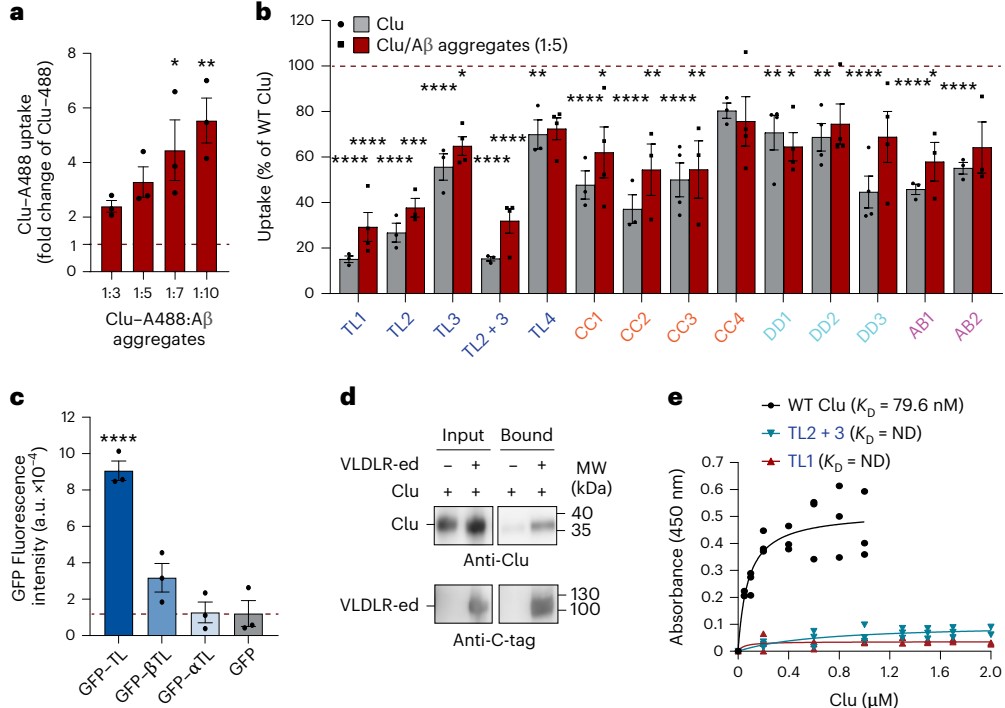

**Fig. 5 | Clu cellular uptake and receptor binding. a,b,** Clu uptake by neuronal cells in the absence or presence of substrate (Aβ aggregates). Fluorescently labeled WT Clu (**a**) or mutants (**b**) (Clu–A488) were added to iNeurons in the absence or presence of Aβ aggregates and uptake was monitored by flow cytometry. **a,** Fold change of WT Clu uptake in the presence of increasing concentration of Aβ aggregates compared to WT Clu alone (molar ratio of Clu/Aβ indicated; dashed line indicates WT Clu uptake in the absence of substrate). Data represent averages ± s.e.m. ($n = 3$ biological replicates). *$P < 0.05$ and **$P < 0.01$ according to one-way ANOVA with Dunnett's post hoc test comparing Clu–A488/Aβ aggregates to Clu–A488 (1:7, $P = 0.0151$; 1:10, $P = 0.0026$). **b,** Percentage uptake of each mutant in the absence (gray) or presence of Aβ aggregates (molar ratio of Clu mutant/Aβ: 1:5; red) relative to WT Clu or WT Clu/Aβ aggregates (dashed line indicates WT Clu or WT Clu/Aβ aggregates). Data represent averages ± s.e.m. (Clu alone, $n = 3$ biological replicates, except for: CC3, DD1, DD2 and DD3, $n = 4$; Clu/Aβ, $n = 4$ biological replicates, except for: TL2/Aβ, CC2/Aβ, CC3/Aβ, AB1/Aβ and AB2/Aβ, $n = 3$). *$P < 0.05$, **$P < 0.01$, ***$P < 0.005$ and ****$P < 0.0001$

according to one-way ANOVA with Dunnett's post hoc test comparing Clu–A488 mutants to WT Clu–A488. **c,** Effect of Clu hydrophobic tails on uptake. GFP–Clu tail fusion proteins (Fig. 4a) were added to iNeurons and uptake was monitored by flow cytometry (a.u., arbitrary units; dashed line indicates GFP uptake). Data represent averages ± s.e.m. ($n = 3$ biological replicates). ****$P = 7.88 \times 10^{-5}$ according to one-way ANOVA with Dunnett's post hoc test comparing GFP–TL to GFP. **d,** Binding of WT Clu to VLDLR detected by immune affinity chromatography. WT Clu and VLDLR-ed were incubated at equimolar concentration in the presence of anti-C-tag resin. WT Clu alone was used as a control. Bound proteins were analyzed by SDS–PAGE and immunoblotting. Representative immunoblots ($n = 3$ biological replicates). MW, molecular weight. **e,** Affinity of WT Clu and selected mutants to immobilized VLDLR-ed. Microtiter plates were coated with VLDLR-ed and incubated with Clu at indicated concentrations. Bound Clu was quantified by ELISA. Background binding to BSA was subtracted. Individual data points are from three biological replicates; binding curves and $K_D$ values are shown. ND, not determined.

Extended Data Fig. 8d). We determined a molar ratio of Clu to DMPC of ~1:100 in these nanodiscs, corresponding to ~9 and 16 Clu molecules per nanodisc particle, respectively, suggesting that Clu may associate with both the rim and the surfaces of the nanodiscs. The Clu molecules in nanodiscs are exchangeable, as judged by the ability of excess unlabeled Clu to displace lipid-bound Clu–A488 (Extended Data Fig. 8e).

The Clu–DMPC nanodisc preparations were nearly as effective as free Clu in preventing D-Rho aggregation (Fig. 6c and Extended Data Fig. 9a) and only slightly less active than free Clu in delaying Aβ aggregation (Fig. 6d and Extended Data Fig. 9b). To determine whether Clu retains chaperone activity in the lipid-bound state, we added nanodiscs to a D-Rho aggregation reaction and then separated them from free Clu by flotation in a density gradient (Extended Data Fig. 9c). A fraction of D-Rho was recovered with the Clu–DMPC complexes, indicating that the Clu-containing nanodiscs participate directly in holdase activity (Extended Data Fig. 9c). To assess the contribution of the Clu hydrophobic tails to chaperone activity when lipid bound, we used limited proteolysis with chymotrypsin, which cleaves after aromatic residues. Analysis of free Clu by MS showed that the β-tail (residues 204–227, mutated region in TL2 mutant) is most sensitive to cleavage, consistent with flexibility and solvent exposure (Extended Data Fig. 9d). Of note, residues 234–240 mapping to the α-tail showed fast H/DX,

indicating flexibility in this region as well (Extended Data Fig. 3a). Incorporation into nanodiscs of WT Clu or the TL4 single-chain mutant had little effect on cleavage by chymotrypsin, suggesting that the β-tail region and possibly the α-tail are solvent exposed in the nanodiscs (Extended Data Fig. 9d,e). Thus, the nanodisc particles are chaperone active, likely using flexible tails of Clu for client binding.

Surprisingly, lipoprotein formation was reduced by ~80–90% in mutants with deleted or changed tail sequences (TL1, TL2, TL3 and TL2 + 3) (Fig. 6e and Extended Data Fig. 9f). It was essentially eliminated with mutant TL2 + 3, in which all hydrophobic residues in the tails are substituted by serine (Fig. 6e and Extended Data Fig. 9f), whereas single-chain Clu (TL4) showed normal lipid binding (Fig. 6e and Extended Data Fig. 9f). Interestingly, lipid binding was also abolished upon deletion of the amphipathic helix α7 (mutant CC1) that follows after the α-tail (mutant TL3) (Fig. 6e and Extended Data Fig. 9f). Of note, in lipoprotein particles containing ApoE or ApoA1, pairs of (albeit longer) amphipathic helices shield the hydrophobic rim of phospholipid discs[48,49]. Using GFP fusion constructs, we found that the α-tail and helix α7 in combination (residues 228–262; GFP–αTL-H7) conferred detectable lipid-binding activity (Fig. 6f), consistent with the mutational data (Fig. 6e and Extended Data Fig. 9f), whereas constructs containing helix α7 (GFP–H7) or the α-tail alone (GFP–αTL) were not

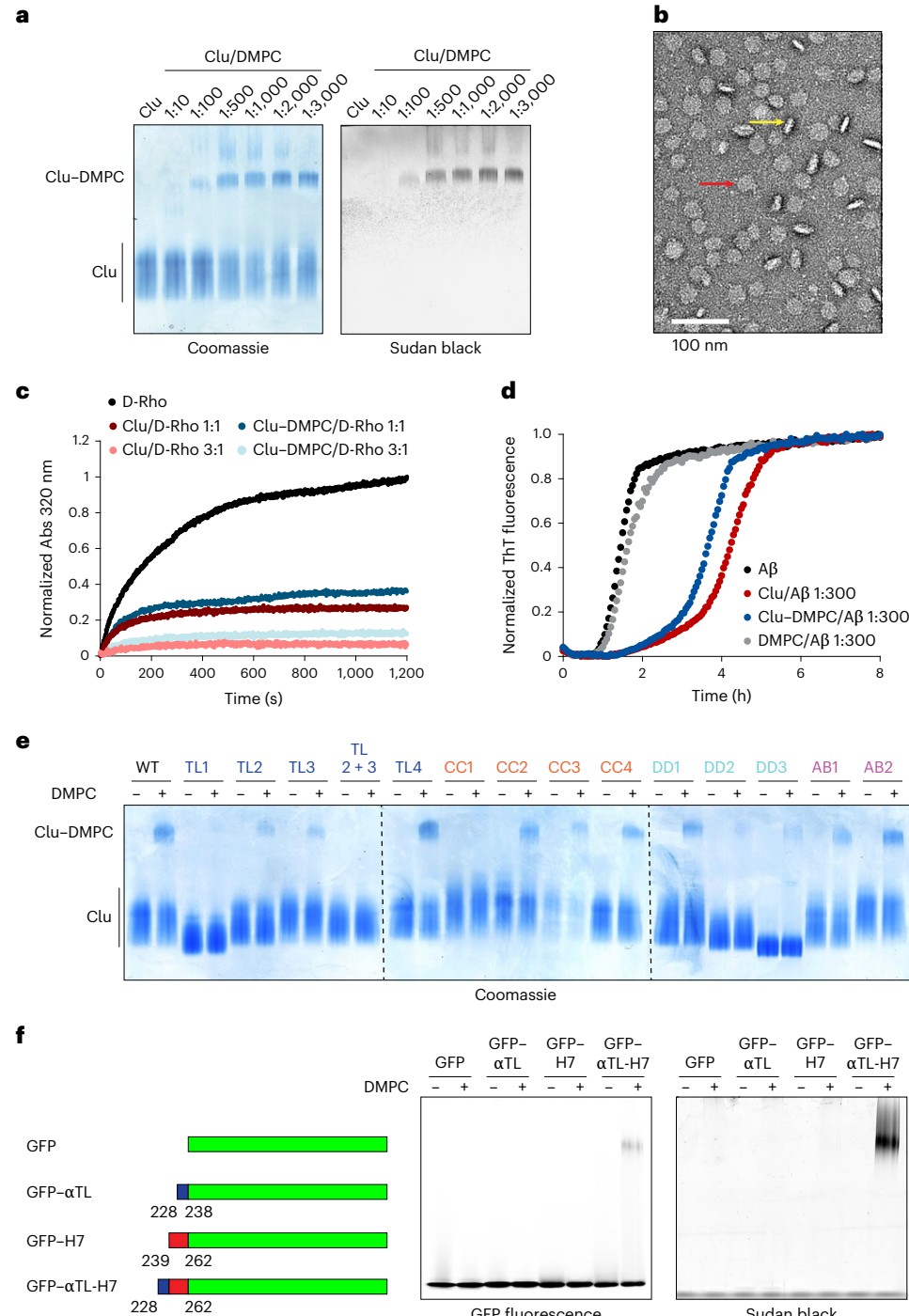

**Fig. 6 | Probing structure and function of Clu lipoprotein complexes.**
**a**, Formation of Clu lipoprotein complexes detected by native PAGE. WT Clu
and DMPC at the indicated molar ratios were cycled above and below the
melt transition temperature of DMPC. Representative native gels stained by
Coomassie blue (protein, left) and Sudan black (lipid, right) are shown (*n* = 3
biological replicates). Lipoprotein complex (Clu–DMPC) and free protein
(Clu) are indicated. **b**, Analysis of Clu lipoprotein complexes by negative-stain
electron microscopy. A representative micrograph is shown (*n* = 3 biological
replicates). Side and top views of nanodiscs are indicated by yellow and red
arrows, respectively. Scale bar, 100 nm. **c**, Effect of Clu lipoprotein complexes
on aggregation of D-Rho. Aggregation assays were performed as in Fig. 3b with
purified Clu lipoprotein complexes at the indicated molar ratios (shades of blue).
Free Clu is shown for comparison (shades of red). Representative absorbance
curves are shown (*n* = 5 biological replicates, except for: Clu/D-Rho 1:1, *n* = 4, and
Clu/D-Rho 3:1, *n* = 3). Quantification is shown in Extended Data Fig. 9a. **d**, Effect

of Clu lipoprotein complexes on Aβ amyloid formation. Amyloid formation was
monitored as in Fig. 3d in the absence (black) or presence of Clu (red), purified
Clu lipoprotein complexes (Clu–DMPC, blue) or DMPC (gray) (molar ratio Clu/
Aβ: 1:300 or corresponding amount of DMPC). Representative normalized traces
are shown (*n* = 4 biological replicates). Quantification is shown in Extended
Data Fig. 9b. **e**, Analysis of structural determinants of Clu lipoprotein complex
formation. Lipoprotein complex formation of Clu mutant proteins with DMPC
at a 1:500 molar ratio was monitored by native PAGE. Representative Coomassie
blue-stained gels are shown (*n* = 3 biological replicates). Quantification is shown
in Extended Data Fig. 9f. **f**, Lipoprotein complex formation with segments of
Clu fused to GFP. Left, domain structure of the GFP fusion proteins. Complex
formation was performed as in **a** at a 1:1,000 molar ratio (construct/DMPC) (*n* = 3
biological replicates). The native PAGE gel was analyzed for GFP fluorescence
(middle) and stained with Sudan black (right). The fluorescence of the free
proteins is saturated.

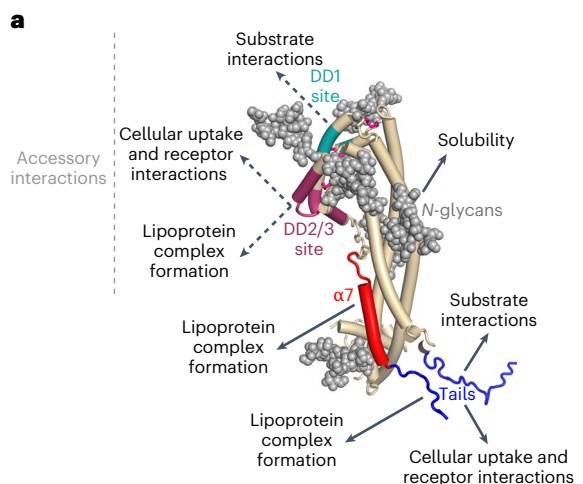

**Fig. 7 | Functional assignment of Clu structural elements. a,** Structural determinants for the functions of extracellular Clu. The respective main and accessory regions for substrate interactions, cellular uptake and lipoprotein complex formation are mapped onto a simplified structure model of Clu. Helices are represented as tubes. Disordered tails (blue), helix α7 (red) and the DD1 (teal) and DD2/3 (purple) sites are indicated. *N*-glycans (gray) are shown in space-filling mode. **b,** Model for cellular uptake of Clu bound to substrate. The disordered tails of Clu presumably interact with the ligand-binding repeats of LDL-type receptors (brown circles; Extended Data Fig. 7c). Formation of Clu dimers or binding of multiple Clu molecules to small aggregates (gray) might enable simultaneous interaction with client protein and cell surface receptors. Client protein-specific receptors might further enhance uptake of Clu–client complexes[27].

sufficient for lipid binding (Fig. 6f). GFP–αTL-H7, similar to full-length Clu, formed disc-shaped particles with DMPC, containing small densities (likely GFP moieties) lining the rims (Extended Data Fig. 9g). In the context of the lipoprotein complex, the α-tail might extend helix α7. Of interest in this context, the combined α-tail and β-tail in GFP–TL failed to mediate lipoprotein complex formation, indicating that the tails alone are not sufficient for this function (Extended Data Fig. 9h). Similar to single-tail mutants (TL2 and TL3), the Clu mutants DD2 and DD3 also showed markedly diminished incorporation into lipid particles (Fig. 6e and Extended Data Fig. 9f). Nanodisc formation, thus, depends on multiple structural elements of Clu, including the flexible tails and the amphipathic helix α7, with DD having an accessory role.

These data are consistent with a model in which the hydrophobic tail sequences initiate lipid binding, followed by undocking of helix α7 from the CC domain and association with the nanodisc rim. The hydrophobic β-tail and possibly the α-tail, while partitioning into the phospholipid bilayer, remain dynamic and accessible for chaperoning misfolded client proteins.

## Discussion

Our structure-based analysis of Clu function provides insight into the mechanism of action of this medically important extracellular chaperone and apolipoprotein[1,50,51]. The crystal structure of human Clu revealed a discontinuous three-domain architecture consisting of a coiled-coil bundle flanked by a conserved helical domain stabilized by disulfide bonds and an α/β roll-like domain (Fig. 2a). Surprisingly, we found two disordered hydrophobic tails, generated by cleavage of the central region of the Clu precursor, to be critical for multiple functionalities, including chaperone activity, binding to cell surface receptors, cellular uptake and lipoprotein complex formation (Fig. 7a).

The hydrophobic tails of Clu resemble the N-terminal substrate-binding regions of intracellular sHsp chaperones in terms of amino acid composition and biophysical properties[37], suggesting convergent evolution of chaperone activity in the phylogenetically younger Clu. The enrichment of these sequences in Clu with phenylalanine, histidine and arginine residues might enable promiscuous π-stacking interactions with non-native substrate proteins and also confer versatility in receptor and lipid interactions. Indeed, deletion of the tails or replacement of their hydrophobic residues with serine resulted in a loss of chaperone function in stabilizing non-native proteins in solution and inhibiting

amyloid formation by Aβ(1–42), α-synuclein and tau. Considering the extensive modification of Clu with hydrophilic *N*-glycans, Clu would be highly efficient in solubilizing aggregation-prone client proteins forming both amorphous or amyloid aggregates, rendering them available for cellular uptake and degradation. As shown with the GFP fusion constructs, each tail in isolation has strongly diminished chaperone activity compared to both tails attached to the same GFP, suggesting synergistic (simultaneous) interactions with substrate proteins. Considering that the tails are critical for client protein and receptor binding, we suggest that formation of Clu dimers (or higher oligomers) ensures that tail sequences are available for both functions (Fig. 7b), although client-specific receptors may also contribute to cellular uptake[27,52]. Clu dimerization occurs at physiological pH without involvement of the tail regions (Extended Data Fig. 1a,e) and dimers are likely stabilized by high local concentrations when multiple Clu molecules interact with small protein aggregates or lipid discs. Our data and a previous report[53] indicate that Clu lipoprotein complexes are almost fully active in chaperoning client proteins. Thus, either both the α-tail and the β-tail become available in the final lipid complex or, if the α-tail and amphiphilic helix α7 form a constitutive structural element, β-tails of adjacent Clu molecules may cooperate in client interactions. The latter scenario might require that Clu dimers have head-to-head parallel structures. Structural information on the Clu dimer will be required to resolve this question.

The tail sequences of Clu cooperate in aggregation prevention with a highly conserved region in the Clu DD. We found this effect to be relevant in the ability of Clu to delay the nucleation-dependent aggregation of Aβ, where Clu is effective at substoichiometric concentrations[19,38]. Interestingly, the DD1 variant of Clu, containing multiple mutations in a highly conserved DD surface patch, was hyperactive in Aβ aggregation prevention in a manner dependent on the hydrophobic tails. It would appear, therefore, that the chaperone activity of Clu has not been optimized in evolution to suppress the formation of Aβ amyloid structure, allowing the DD to function more broadly in interacting with non-native client proteins and with membrane receptors. Indeed, the DD1 mutant was markedly impaired in D-Rho aggregation prevention, while the DD2 and DD3 mutants showed reduced VLDLR binding. Our findings, thus, demonstrate that Clu can be optimized for aggregation inhibition of specific client proteins. It will be of interest to explore the therapeutic potential of the DD1 mutant and other Clu variants in mouse models of AD.

## Online content

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

## Methods

### Plasmids

The pB-T-PAF-Clu plasmid used for WT Clu expression and purification was constructed by amplifying the coding region of pB-T-PAF-CluStrep[22] by PCR. The PCR product was then digested with NheI and NotI and subcloned into the pB-T-PAF vector. All pB-T-PAF plasmids encoding Clu variants are based on this vector. pB-T-PAF-Clu-TL1 was obtained by PCR amplification of the entire plasmid with the mutagenic primers using Herculase II (Agilent), followed by ligation with the KLD enzyme mix (New England Biolabs). pB-T-PAF-Clu-CC1 and pB-T-PAF-Clu-TL4 were obtained by separate PCR amplification of 5′ and 3′ Clu fragments with the mutagenic primers, followed by digestion with NheI and NotI and simultaneous subcloning into pB-T-PAF. pB-T-PAF-Clu-TL2, pB-T-PAF-Clu-TL3, pB-T-PAF-Clu-TL2 + 3, pB-T-PAF-Clu-CC2, pB-T-PAF-Clu-CC3, pB-T-PAF-Clu-DD1, pB-T-PAF-Clu-DD2, pB-T-PAF-Clu-DD3, pB-T-PAF-Clu-AB1 and pB-T-PAF-Clu-AB2 were obtained by plasmid digestion with NheI and NotI or PCR amplification of the plasmid with primers encoding overlapping regions with the corresponding synthesized gene fragment including the mutations (Twist Bioscience) and assembly with the NEBuilder HiFi DNA assembly master mix (New England Biolabs). pB-T-PAF-Clu-CC4 was obtained by mutagenesis using the Q5 site-directed mutagenesis kit (New England Biolabs). pB-T-PAF-Clu-DD1 + TL1 was obtained by PCR amplification of pB-T-PAF-Clu-DD1 plasmid to delete the TL1 region and assembly with the NEBuilder HiFi DNA assembly master mix (New England Biolabs).

pB-T-PAF-VLDLR(1–797)Δ(751–779)-C-tag was obtained by PCR amplification of VLDLR from human placenta complementary DNA (Edge Biosystems) using Herculase II Fusion DNA polymerase (Agilent Technologies), followed by Gibson assembly with pB-T-PAF using NEBuilder HiFi DNA assembly master mix (New England Biolabs).

pET-Sac-Aβ(M1–42) plasmid for Aβ(M1–42) expression and purification was a gift from D. Walsh (Addgene, plasmid 71875)[54]. pQTEV-LRPAP1 plasmid for RAP expression and purification was a gift from K. Büssow (Addgene, plasmid 31327)[55]. The signal peptide was removed (amino acids 1–35) by mutagenesis using the Q5 site-directed mutagenesis kit (New England Biolabs).

pHUE-eGFP-Clu-tail was generated by restriction and insertion cloning with a DNA fragment encoding eGFP–Clu-tail amplified from pEGFP-C2 (Clontech) by nested PCR using primers encoding Clu(228–238) and Clu(204–227) and introducing SacII and HindIII restriction sites and the SacII–HindIII restriction fragment of plasmid pHUE[56].

pHUE-Clu-αTL-H7-eGFP and pHUE-Clu-H7-eGFP were obtained by PCR amplification and assembly with the pHUE-eGFP-Clu-tail vector backbone digested with SacII and HindIII using the NEBuilder HiFi DNA assembly master mix (New England Biolabs).

pHUE-Clu-αTL-eGFP and pHUE-Clu-βTL-eGFP vectors were generated by restriction and insertion cloning with a DNA fragment encoding the respective insert amplified from pHUE-eGFP-Clu-tail by PCR using primers introducing SacII and HindIII restriction sites and the SacII–HindIII restriction fragment of plasmid pHUE.

A detailed Gibson assembly protocol (NEBuilder HiFi DNA assembly master mix, New England Biolabs) is available from protocols.io (https://doi.org/10.17504/protocols.io.ewov1de6yvr2/v1)[57].

All newly made plasmids were deposited in Addgene (RRIDs available in Supplementary Table 1).

### Cell lines

HEK293-EBNA (HEK293E, CVCL_6974) suspension cell lines[58] stably expressing the recombinant Clu and VLDLR protein constructs (Supplementary Table 1) were generated using a piggyBac transposon-based expression system[59] with the respective pB-T-PAF-Clu plasmids. Newly generated cell lines were registered at Cellosaurus (RRIDs available in Supplementary Table 1).

iPS cell line HPSI0214i-kucg_2 (RRID:CVCL_AE60) was purchased from the UK Health Security Agency (77650065, supplied by HipSci).

iPS cells were maintained at 37 °C and 5% $CO_2$ in mTeSR or mTeSR plus medium (Stem Cell Technologies) on Geltrex-coated (Thermo Fisher Scientific) cell culture plates. Cells were split when confluent using ReLeSR (Stem Cell Technologies). Quality control tests are specified in Supplementary Table 2.

Neural progenitor cells (NPCs) were generated using the STEMdif SMADi neural induction kit (Stem Cell Technologies) following the monolayer protocol. NPCs were frozen in STEMdif neural induction medium with a SMAD inhibitor and 10% DMSO.

iPS cell-derived forebrain-type neurons were generated from the NPCs described above. NPCs were thawed in STEMdif neural induction medium with a SMAD inhibitor on 0.01% poly(L-ornithine) cell culture plates coated with 10–20 µg ml⁻¹ laminin (Merck) and differentiated using the STEMdiff forebrain neuron differentiation kit (Stem Cell Technologies) followed by the STEMdiff forebrain neuron maturation kit (Stem Cells Technologies). STEMdiff forebrain neuron maturation medium was supplemented for 6 days with 5 µM 5-fluorouracil and uridine (Merck) to stop the growth of nondifferentiated cells. After 8 days in STEMdiff forebrain neuron maturation medium (half medium change every other day), iPS cell-derived neurons were maintained in Neurobasal Plus medium (Thermo Fisher Scientific) supplemented with B-27 Plus Supplement 1× (Thermo Fisher Scientific), 0.5 mM GlutaMAX (Thermo Fisher Scientific), 100 U per ml penicillin and 100 µg ml⁻¹ streptomycin sulfate (Thermo Fisher Scientific). A detailed differentiation protocol is available from protocols.io (https://doi.org/10.17504/protocols.io.eq2lyxz3wgx9/v1)[60].

### Protein expression and purification

All protein purification steps were performed at 4 °C unless otherwise indicated. Protein concentrations in the final preparations were determined by absorbance at 280 nm using absorbance coefficients calculated from the protein sequence with the program ProtParam (Swiss Institute of Bioinformatics) unless otherwise noted. Purified protein samples were concentrated by ultrafiltration and snap-frozen in liquid nitrogen for storage at −70 °C.

### Clu constructs

Clu constructs were expressed and secreted by the respective HEK293E stable cell lines cultured in FreeStyle 293 expression medium (Thermo Fisher Scientific) for 4 days. For expression of Clu constructs containing oligomannose N-glycans, the α-mannosidase I inhibitor kifunensine (MedChemExpress) dissolved in water was added to the medium (16 µM final concentration). The conditioned medium was then separated from the cells by centrifugation. For chromatographic purification, 200 ml of medium was first dialyzed against wash buffer (20 mM sodium acetate pH 5.0). After removal of precipitate by centrifugation, the supernatant was passed over HiTrap SP XL cation-exchange resin (Cytiva, stack of five 5 ml columns). The column stack was washed with five column volumes of wash buffer. For protein elution, a 0–500 mM NaCl gradient in wash buffer was applied. Clu-containing fractions were further purified by SEC on a HiLoad 26/600 Superdex-200 (Cytiva) in 20 mM sodium acetate pH 5.0, 100 mM NaCl and 1 mM EDTA. Fractions containing pure, monomeric Clu were merged. A detailed protocol, however with an additional denaturing wash, is available from protocols.io (https://doi.org/10.17504/protocols.io.bvvkn64w)[61].

### VLDLR-ed

VLDLR(28–797)Δ(751–779)-C-tag (VLDLR-ed, derived from UniProt P98155-2) was expressed and secreted by HEK293-VLDLR(1–797)Δ(751–779)-C-tag cells cultured in FreeStyle 293 expression medium (Thermo Fisher Scientific) for 4 days. The conditioned medium was then separated from the cells by centrifugation. For chromatographic purification, the medium was first dialyzed against binding buffer (20 mM Tris-HCl pH 7.2, 100 mM NaCl and 0.5 mM $CaCl_2$) overnight.

VLDLR-ed in the dialysate was purified by affinity chromatography using CaptureSelect C-tag affinity matrix (Thermo Scientific) using an elution buffer containing 20 mM Tris-HCl pH 7.0, 2 M MgCl₂ and 2 mM CaCl₂. Finally, the protein buffer was exchanged with an NAP-25 desalting column (Cytiva) equilibrated in binding buffer. A detailed protocol is available from protocols.io (https://doi.org/10.17504/protocols.io.j8nlkoqw1v5r/v1)[62].

## eGFP–Clu constructs

eGFP–Clu constructs were expressed as C-terminal His₆–ubiquitin fusion proteins in *Escherichia coli* Bl21-CodonPlus (DE3)-RIL cells (Agilent) transformed with the corresponding plasmid and cultured in 4 L of Luria–Bertani medium overnight at 18 °C with 0.25 mM IPTG. After lysis of the cells by ultrasonication on ice in PBS buffer containing 20 mM imidazole, 1 mM PMSF and cOmplete EDTA-free protease inhibitor cocktail (Roche), the clarified supernatant was subjected to affinity chromatography on Ni-chelating Sepharose (Cytiva). Green-fluorescent elution fractions were merged and digested with His-tagged Usp2 for 4 h on ice. After buffer exchange into PBS buffer containing 20 mM imidazole using a HiPrep 26/10 desalting column (Cytiva), a second affinity chromatography on Ni-chelating Sepharose (Cytiva) was performed. The green-fluorescent fractions of unbound protein were pooled and concentrated by ultrafiltration and the concentrate subjected to SEC on a HiLoad 16/600 Superdex-200 (Cytiva) column equilibrated with PBS. Concentrations were determined by absorbance at 488 nm using an absorbance coefficient of 61,000 M⁻¹ cm⁻¹. A detailed protocol is available from protocols.io (https://doi.org/10.17504/protocols.io.n2bvj3w55lk5/v1)[63].

## Aβ

Aβ(M1–42) was purified as described previously[64] with modifications. Aβ(M1–42) was expressed in *E. coli* BL21(DE3) cells transformed with the pET-Sac-Aβ(M1–42) plasmid by IPTG induction. The cell pellet from a 3-L culture was resuspended in 80 ml of TE buffer pH 7.5 (10 mM Tris-HCl pH 7.5 and 1 mM EDTA) supplemented with cOmplete EDTA-free protease inhibitor cocktail (Merck) and 2.5 U per ml benzonase until the sample was homogeneous. Cells were lysed by sonication and the lysate was cleared by centrifugation (10 min, 18,000*g* at 4 °C). The pellet was resuspended in 50 ml of TE buffer pH 7.5 until the sample was homogeneous, sonicated and centrifuged (10 min, 18,000*g* at 4 °C). This step was repeated twice. The pellet was then resuspended in 40 ml of TE buffer pH 9.5 until the sample was homogeneous, sonicated and centrifuged (10 min, 18,000*g* at 4 °C). The supernatant was collected and filtered through a 0.45-µm filter; the pH was adjusted to 8.5 using 1 M HCl. The sample was then loaded onto a DEAE cellulose column equilibrated with TE buffer pH 8.5. The column was washed with TE buffer pH 8.5 and 10 mM NaCl and eluted with TE buffer pH 8.5 and 50 mM NaCl. Fractions containing pure Aβ peptide were combined and the buffer was exchanged to TE buffer pH 8.5 using a desalting column. Then, a 100-ml elution from the desalting column was frozen in liquid N₂ while rotating and lyophilized. The lyophilized sample was resuspended in 15 ml of GE buffer pH 8.5 (6 M guanidinium hydrochloride and 20 mM sodium phosphate) and loaded onto a Superdex 75 SEC column equilibrated with AE buffer pH 8.5 (20 mM ammonium acetate and 0.2 mM EDTA). Fractions containing monomeric Aβ were combined and peptide concentration was determined by absorbance at 205 nm. Finally, 500-µg aliquots were flash-frozen, lyophilized and stored at −70 °C. A detailed protocol is available from protocols.io (https://doi.org/10.17504/protocols.io.3byl4z2k2vo5/v1)[65].

## Crystallization

Crystals of Clu mutant Clu-Δ(214–238) (TL1) with oligomannose *N*-glycans were obtained with the help of the Max Planck Institute of Biochemistry (MPIB) Crystallization Facility by the sitting-drop vapor diffusion method using the Classics (Qiagen), Index (Hampton Research) and JCSG-plus (Molecular Dimensions) crystallization screens at 4 °C by mixing 200 nl of sample with 200 nl of reservoir.

Crystal form I of space group *P*2₁ was obtained with 30% PEG-4000, 0.2 M ammonium acetate and 0.1 M sodium citrate pH 5.6 (Classics condition H3), 0.05 M ammonium sulfate, 0.05 M Bis–Tris pH 6.5 and 30% v/v pentaerythritol ethoxylate (15/4 EO/OH) (Index condition E9) and 0.2 M lithium sulfate monohydrate, 0.1 M Bis–Tris pH 6.5 and 25% PEG-3350 (Index condition G3) as precipitant. For cryoprotection, precipitants additionally containing 7.5% and 15% glycerol were prepared. The crystals were serially incubated with these buffers for each 20 min before flash-cooling in liquid nitrogen. Diffraction data from the first condition were used for model building and refinement.

Crystal form II of space group *C*2 was obtained with 24% PEG-1500 and 15% glycerol (JCSG-plus condition D1). These crystals were directly flash-cooled in liquid nitrogen. A detailed protocol is available from protocols.io (https://doi.org/10.17504/protocols.io.bp2l68kndgqe/v1)[66].

## Structure solution and refinement

X-ray diffraction data were collected at 100 K and a wavelength of 0.7749 Å by the oscillation method at beamline ID23-1 at the European Synchrotron Radiation Facility (ESRF).

The data were integrated and scaled with XDS (Version Feb 5, 2021; https://xds.mr.mpg.de/)[67]. The programs Pointless (version 1.12.4)[68], Aimless (version 0.7.4)[69] and Ctruncate (version 1.17.29)[70], as implemented in the CCP4i graphical user interface (version 7.1.010; https://www.ccp4.ac.uk/)[71], were used for data reduction. Partial datasets collected from different portions of the crystals were merged. The X-ray diffraction was strongly anisotropic and an overall *I/σI* of 1.5 in the outer resolution shell was used as resolution cutoff criterion.

The crystal structure of Clu mutant TL1 (Clu-Δ(214–238)) in space group *P*2₁ was solved by molecular replacement with Molrep (version 11.4.06; CCP4i interface version 7.1.010)[72] using a truncated version of the AlphaFold2 model for human Clu (https://alphafold.ebi.ac.uk/entry/P10909)[73,74] in which low-confidence regions were trimmed away. WinCoot (version 0.9.4.1; http://bernhardcl.github.io/coot) was used for manual model building[75]. The model was refined with REFMAC5 (version 5.8.0267; CCP4i interface 7.1.010)[76]. The final refinement was performed with phenix.refine (version 1.19.2-4158; http://www.phenix-online.org/)[77]. Residues facing solvent channels with disordered side chains were truncated after Cβ. The final model contains one copy of Clu mutant TL1 per asymmetric unit, with residues 23–24, 261–280 and 447–449 missing for lack of discernible electron density, probably because of disorder. The model includes partial structures for the oligomannose *N*-glycans attached to N86 (GlcNAc), N103 (GlcNAc-GlcNAc), N145 (Man-GlcNAc-GlcNAc), N291 (GlcNAc), N354 (GlcNAc) and N374 (GlcNAc) of Clu mutant TL1. The model has excellent stereochemistry with 98.0% of the residues in the favored regions of the Ramachandran plot and no outliers.

The crystal structure of Clu mutant TL1 (Clu-Δ(214–238)) in space group *C*2 was solved by molecular replacement with Molrep (version 11.4.06)[72] using a preliminary model for Clu mutant TL1 in the *P*2₁ crystal lattice. WinCoot (version 0.9.4.1) was used for manual model building[75]. The model was refined with REFMAC5 (version 5.8.0267) using local noncrystallographic symmetry restraints[76]. The final refinement was performed with phenix.refine (version 1.19.2-4158)[77]. Residues facing solvent channels with disordered side chains were truncated after Cβ. The final model contains two copies of Clu mutant TL1 per asymmetric unit. Residues 23–24, 197–211, 261–279 and 449 in chain A and residues 195–211, 260–279 and 446–449 in chain D are missing for lack of discernible electron density, probably because of disorder. The model includes partial structures for the oligomannose *N*-glycans attached to N86, N103, N145, N291, N354 and N374 in chains A and D of TL1. The model has reasonable stereochemistry with 95.2% of the residues in the favored regions of the Ramachandran plot and 0.28% outliers.

## Structure analysis

Coordinates were aligned with Lsqkab (version 7.1.010; CCP4i interface 7.1.010) and Lsqman (version 081126/9.7.9)[78]. Molecular drawings and sequence alignment depictions were generated with the programs Pymol (version 2.2.3; http://www.pymol.org) and ESPript (version 3.0; https://espript.ibcp.fr)[79], respectively.

## Sequence data analysis

An alignment of 242 representative Clu sequences was created with the Consurf server (https://consurf.tau.ac.il/consurf_index.php)[80] using human Clu as input with the default settings. The amino acid frequency of sequences corresponding to residues 204–238 was analyzed. Sequences for human α-crystallin A and B homologs in jawed vertebrates (taxonomy ID 7776) were retrieved from UniProt using BLAST (https://www.uniprot.org/blast), yielding 662 CRYAA/HSPB4 and 444 CRYAB/HSPB5 sequences, respectively, and aligned with Clustalo (https://www.ebi.ac.uk/jdispatcher/msa/clustalo). The NTDs (that is, the sequences before the α-crystallin domain, which begins at residue 62 in human HSPB4 and residue 65 in human HSPB5) were cut from the alignments and analyzed. As reference data, the reviewed proteome sequences of *Homo sapiens* and the predicted proteome sequences of *Callorhinchus milii* (taxonomy ID 7868), *Latimeria chalumnae* (taxonomy ID 7897), *Danio rerio* (taxonomy ID 7955), *Xenopus laevis* (taxonomy ID 8355), *Gallus gallus* (taxonomy ID 9031), *Ornithorhynchus anatinus* (taxonomy ID 9258) and *Monodelphis domestica* (taxonomy ID 13616) from UniProt were used. The sequence information is provided in the Source Data. The amino acid frequency analysis was performed with Excel (version 16.0.14332.20788).

## Protein aggregation reactions and thioflavin T (ThT) fluorescence measurements

**Rho aggregation assay.** Bovine rhodanese (Rho) (60 μM) was denatured in 6 M guanidinium–HCl and 5 mM DTT for 1 h at 25 °C and diluted 120-fold into PBS pH 7.2 (Gibco), PBS pH 7.4 (in house) or 20 mM sodium acetate pH 5.2, 150 mM NaCl and 2 mM $CaCl_2$ (final concentration 0.5 μM) in the absence or presence of Clu constructs (0.5 μM or 1.5 μM). Aggregation was monitored immediately after dilution at 25 °C by measuring turbidity at 320-nm wavelength using a Jasco V-560 spectrophotometer with the V-500 Control Driver (version 1.41.02). The data were fitted using Sigma Plot (version 14.0; https://grafiti.com/sigmaplot-detail/) (exponential rise to maximum, double, four-parameter function) to obtain the maximum plateau absorbance. A detailed protocol is available from protocols.io (https://doi.org/10.17504/protocols.io.j8nlkdbrxg5r/v1)[81].

**Aβ(M1–42) aggregation.** Aβ aggregation assays were performed as previously described[64]. Briefly, 500 μg of lyophilized Aβ was resuspended in 700 μl of GE buffer pH 8.5 and loaded onto a Superdex 75 SEC column (GE Healthcare) equilibrated with aggregation buffer (20 mM sodium phosphate and 0.2 mM EDTA, pH 8.5). Monomeric Aβ was collected in low-binding tubes and the concentration was estimated by absorbance at 205 nm. Aβ monomers were diluted to 4 μM in aggregation buffer containing 10 μM ThT in the absence or presence of WT Clu or Clu mutant proteins. Then, 80 μl of the mix was dispensed per well in a 96-well half-area plate of black polystyrene with a clear bottom (Corning, 3881). Samples were measured in quadruplicate in each plate (technical replicates). The ThT signal (excitation 440 nm, emission 480 nm) was measured every 3 min in a CLARIOstar plate reader (BMG Labtech) with the ClarioStar software (version 5.70 R3) at 37 °C. The data were extracted using the MARS software (version 4.01 R2) and fitted using Sigma Plot (version 14.0; https://grafiti.com/sigmaplot-detail/) (sigmoidal, sigmoid, three-parameter function) to obtain the half-time for reaching the aggregation plateau. A detailed protocol is available from protocols.io (https://doi.org/10.17504/protocols.io.bp2l68ky5gqe/v1)[82].

## Protein labeling

Clu was labeled with Alexa488 (A488) *N*-hydroxysuccinimide ester (Thermo Fisher Scientific). Before the labeling reaction, the buffer was exchanged with 0.1 M sodium bicarbonate buffer pH 8.3 (N-terminal labeling buffer) using a Nap5 column and labeling was subsequently performed at a fourfold molar excess of A488 for 1.5 h at room temperature. Free dye was removed using a Nap5 column, pre-equilibrated with PBS buffer. The labeling efficiency was measured by nanodrop and was typically about 70–180% (note that Clu contains two N termini). A detailed protocol is available from protocols.io (https://doi.org/10.17504/protocols.io.rm7vzxpbrgx1/v1)[83].

## Cellular uptake assays

First, 5 μg ml$^{-1}$ Clu–A488 or GFP construct in 400 μl of medium (200 μl of fresh medium) was added to 250,000 iNeurons cultured in a well of a 12-well plate. After 1 h, cells were placed on ice, washed with PBS and collected with Accutase (Stem Cell technologies). Cells were washed once with PBS, fixed with 4% PFA in PBS for 10 min, washed with PBS, resuspended in 160 μl of PBS and stored at 4 °C until analysis. For measuring Clu–A488 uptake in the presence of substrate, 1 μM Clu–A488 was incubated with the corresponding amount of Aβ aggregates for 20 min at 37 °C (total volume: 40 μl in PBS). After incubation, the mix was diluted in 400 μl of medium (200 μl of fresh medium; Clu–A488 final concentration: 5 μg ml$^{-1}$). Cells were analyzed at the MPIB Imaging Facility with an Attune NxT flow cytometer with the Attune NxT software (version 5.1.1; Thermo Fisher Scientific). Right before measuring, 50 μl of Trypan blue solution 0.4% (Thermo Fisher Scientific) was added to each sample to quench the A488 fluorescence outside the cells. Uptake was recorded during the linear increase of the signal (that is, before degradation became apparent). To measure the A488 or GFP signal, cells were excited with 488-nm laser light and fluorescence was determined using the 530/30 filter. For each sample, at least 10,000 cells were analyzed (average number of analyzed cells: ~140,000). Data processing was performed using MatLabR2021b (https://github.com/csitron/MATLAB-Programs-for-Flow-Cytometry). Cells were gated by size using forward scatter (FSC-H) (Supplementary Fig. 1) and the A488 mean intensity normalized by FSC-H of each mutant was normalized by its own labeling efficiency. A detailed protocol is available from protocols.io (https://doi.org/10.17504/protocols.io.14egn3k2yl5d/v1)[84].

## Clu–VLDLR-ed binding assay

WT Clu in the presence or absence of VLDLR-ed, all at 5 μM, was incubated with 50 μl CaptureSelect C-tag affinity resin (Thermo Fisher Scientific) in C-tag wash buffer (20 mM Tris-HCl pH 7.4, 100 mM NaCl and 2 mM $CaCl_2$) for 2 h at 25 °C, followed by transfer into spin columns (MoBiTec). Subsequently, the gel bed was washed four times with 100 μl of C-tag wash buffer. Bound protein was eluted with three times 50 μl of C-tag elution buffer (20 mM Tris-HCl pH 7.0, 2 M $MgCl_2$ and 2 mM $CaCl_2$). Protein association was analyzed by SDS–PAGE and immunoblotting against Clu α-chain and C-tag. A detailed protocol is available from protocols.io (https://doi.org/10.17504/protocols.io.4r3l26m2qv1y/v1)[85].

## Solid-phase binding assay

The assay was performed as previously described[41]. The 96-well plate (Nunc-immuno MicroWell 96-well solid plate; Merck) was coated with 100 μl of TBS-C (Tris-buffered saline pH 7.4 and 2 mM $CaCl_2$) containing 10 μg ml$^{-1}$ VLDLR-ed overnight at 4 °C. The plate was washed once with TBS-C and incubated for 2 h at room temperature with TBS-C blocking buffer (2% BSA and 0.05% Tween-20). The plate was then incubated with different concentrations of the ligands Clu or RAP in blocking solution for 1 h at room temperature. For the competition assay, the plate was incubated with 100 nM of Clu in the presence of increasing concentrations of RAP. The plate was washed three times with blocking solution. Anti-Clu (sc-5289, Santa Cruz Biotechnologies; dilution 1:100) or anti-RAP (sc-515625, Santa Cruz Biotechnologies;

dilution 1:100) antibodies were added in TBS-C blocking solution and incubated for 1 h at room temperature. The plate was washed three times with TBS-C blocking solution and incubated with horseradish peroxidase (HRP)-conjugated goat anti-mouse IgG (A4416, Merck; dilution 1:10,000) added in TBS-C blocking solution and incubated for 1 h at room temperature. The plate was washed three times with TBS-C blocking solution and developed by adding 100 μl per well of the HRP substrate one-step Ultra TMB ELISA substrate solution (Thermo Fisher Scientific) and the reaction was quenched with 100 μl per well of 2 M sulfuric acid. Absorbance at 450 nm was measured in a Spark multimode microplate reader (Tecan). Binding to wells coated with BSA was used to estimate the background signal for each sample and the no-ligand well signal was subtracted from the rest of the samples. The $K_D$ values were obtained by fitting the data using GraphPad Prism 10 software (www.graphpad.com) (binding, saturation, one-site-specific binding). A detailed protocol is available from protocols.io (https://doi.org/10.17504/protocols.io.yxmvm36kol3p/v1)[86].

#### Formation and isolation of protein–phospholipid particles

First, DMPC powder (Avanti Polar Lipids, Sigma) was dissolved in 3:1 chloroform and methanol at 25 mg ml$^{-1}$ or 5 mg ml$^{-1}$ and stored at −70 °C as stock solution. For the formation of Clu–phospholipid particles, the required amount of DMPC was transferred to a glass vial and solvent was removed by evaporation through a constant stream of nitrogen gas. The dried film was resuspended in 1× PBS pH 7.2 (Gibco), vortexed and sonicated in a Bioruptor sonication bath (Diagenode) (25 cycles of 5 s on, 5 s off). Then, 20 μM Clu, Hsc70, ApoA1 (CYT-661, Prospec, Hölzel Diagnostik), the respective Clu–GFP construct or 6 μM ApoE (CYT-874, Prospec, Hölzel Diagnostik) was mixed (1:1, v/v) at the indicated ratios with DMPC in a total volume of 100 μl. For DMPC–Clu lipoprotein complex isolation by SEC, 20 μM Clu or Clu–A488 was mixed with 20 or 10 mM DMPC. The sample was incubated through three cycles of 18 °C for 15 min and 30 °C for 15 min, as described previously[28]. The formation of the Clu–phospholipid particles was analyzed by native PAGE. Coomassie blue protein staining was performed with InstantBlue (Abcam) or with 0.1% (w/v) Serva blue R in 10% acetic acid and 50% ethanol followed by destaining with 10% acetic acid and 10% ethanol. The lipoprotein complex band formed in presence of Clu mutant proteins was quantified by densitometry with Image J. The detection of lipids after native PAGE was performed by incubating the gel overnight with freshly prepared 0.4% Sudan black B (Merck) in 16.7% acetone and 12.5% acetic acid solution followed by destaining with 20% acetone and 15% acetic acid. When Clu–A488 or the GFP constructs were used, the fluorescence signal of the native PAGE gel was analyzed using a Typhoon 5 imager (Cytiva) with the Amersham Typhoon control software (version 2.0.0.6).

When Clu–phospholipid particles were isolated by SEC, after lipidation, the sample was briefly centrifuged to remove large particles and the supernatant was loaded onto a Superose-6 column equilibrated with 1× PBS or TBS-C (for chymotrypsin digestion). After analysis by native PAGE, fractions containing the Clu–DMPC nanodisc particles were concentrated by ultrafiltration using Vivaspin (molecular weight cutoff: 30,000 or 10,000 kDa; GE Healthcare) centrifugal concentrators. Protein concentration was determined by absorbance at 280 nm or with the Bradford assay and lipid content was analyzed by MS (described in Supplementary Information) and with the phospholipid assay kit (MAK122, Merck). A detailed protocol is available from protocols.io (https://doi.org/10.17504/protocols.io.bp2l6x59zlqe/v1)[87].

#### Negative-stain transmission electron microscopy

For negative-stain analysis, continuous carbon grids (Quantfoil) were glow-discharged using a plasma cleaner (PDC-3XG, Harrick) for 30 s. Grids were incubated for 1 min with 4 μl of the fractions of the Clu–DMPC band from SEC (in PBS buffer) or the GFP–αTL-H7 lipoprotein complex preparation, blotted and stained with 2% uranyl acetate solution (Electron Microscopy Sciences), dried and imaged at the MPIB Cryo-Electron Microscopy Core Facility on a Titan Halo (FEI) transmission electron microscope using SerialEM (version 4.1.6). A detailed protocol is available from protocols.io (https://doi.org/10.17504/protocols.io.4r3l29x1jv1y/v1)[88].

#### Statistical analysis

Statistical analysis was performed with GraphPrism 10 (www.graphpad.com). The sample sizes given in the figure legends describe measurements taken from distinct, biological replicates. One-way analysis of variance (ANOVA) with Dunnett's post hoc test was used for multiple comparisons. Exact $P$ values are indicated in the Source Data.

Further methods (protein expression and purification of TauRD, α-Synuclein(A53T), Hsc70 and RAP, gel electrophoresis and immunoblotting, removal of $N$-glycans with PNGase F under denaturing conditions, H/DX–MS, CD spectroscopy, protein aggregation reactions and ThT fluorescence measurements of TauRD and α-synuclein, analytical SEC, immunofluorescence microscopy, MS GFP–VLDLR-ed binding assay, lipidomics, optiprep gradient, limited proteolysis with chymotrypsin and MS proteomics) are provided in the Supplementary Information.

#### Reporting summary

Further information on research design is available in the Nature Portfolio Reporting Summary linked to this article.

### Data availability

The datasets, software, code, protocols and lab materials used and/or generated in this study are listed in Supplementary Table 1, alongside their persistent identifiers on Zenodo (https://doi.org/10.5281/zenodo.14243720)[89]. The ESRF diffraction data can be obtained online (https://doi.org/10.15151/ESRF-ES-541098252)[90]. The coordinates and structure factors reported in this manuscript were deposited to the Protein Data Bank under accession codes 7ZET and 7ZEU. The MS data were deposited to the ProteomeXchange Consortium through the PRIDE partner repository with the dataset identifiers PXD056940 and PXD057022. Source data are provided with this paper.

### Code availability

MatLabR2021b code for flow cytometry data processing can be found on GitHub (https://github.com/csitron/MATLAB-Programs-for-Flow-Cytometry).

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

## Acknowledgements

We thank T. F. Schaller and D. Balchin for their contributions to the early phase of the Clu structural project, T. Klaubert, L. Dinata, G.-V. Datcu, J. Samiee, A. E. Agaciklar and D. Rege for their help in the preparation and characterization of Clu mutants and GFP fusion proteins, N. Wischnewski and S. Gärtner for purification of Aβ(M1–42), Hsc70 and RAP proteins, R. Körner and A. Ries for help with MS experiments, K. Zang for sharing pure eGFP protein, C. Sitron for sharing the flow cytometry analysis code and for helpful discussions and D. Paquet and his team for kindly sharing their expertise on iPS cell culture and differentiation protocols. We acknowledge the ESRF for provision of the macromolecular crystallography infrastructure under proposal number MX-2377 and we would like to thank D. de Sanctis for assistance in using beamline ID23-1. We thank the staff of the MPIB Core (RRID:SCR_025743), Protein Production (RRID:SCR_025741), Cryo-Electron Microscopy (RRID:SCR_025744) and Crystallization (RRID: SCR_025740) facilities, especially J. Scholz, for generating the HEK293-EBNA cell lines and expressing the Clu and VLDLR-ed constructs, M. Spitaler, M. Oster and G. Cardone from the MPIB Imaging facility (RRID:SCR_025739) for support with flow cytometry, imaging and image processing and B. Steigenberger and her team from the MPIB MS core facility (RRID:SCR_025745) for MS lipidomics and proteomics analysis. This research was funded in part by the Deutsche Forschungsgemeinschaft (German Research Foundation) under Germany's Excellence Strategy within the framework of the Munich Cluster for Systems Neurology (EXC 2145 SyNergy, ID: 390857198), the Aligning Science Across Parkinson's initiative (ASAP-000282) through the Michael J. Fox Foundation for Parkinson's Research and the Max Planck Foundation. For the

## Author contributions

A.B. determined and analyzed the Clu structure and designed the mutant constructs. P.Y. designed, performed and analyzed the biochemical experiments with support from A.B. and S.P. A.I.C. performed the H/DX experiments and MS analysis. C.M. performed the negative-stain electron microscopy. A.B., P.Y. and F.U.H. designed the project and wrote the manuscript with input from the other coauthors.

## Funding

## Competing interests

The authors declare no competing interests.

## Additional information

**Extended data** is available for this paper at https://doi.org/10.1038/s41594-025-01631-4.

**Correspondence and requests for materials** should be addressed to Patricia Yuste-Checa, F. Ulrich Hartl or Andreas Bracher.

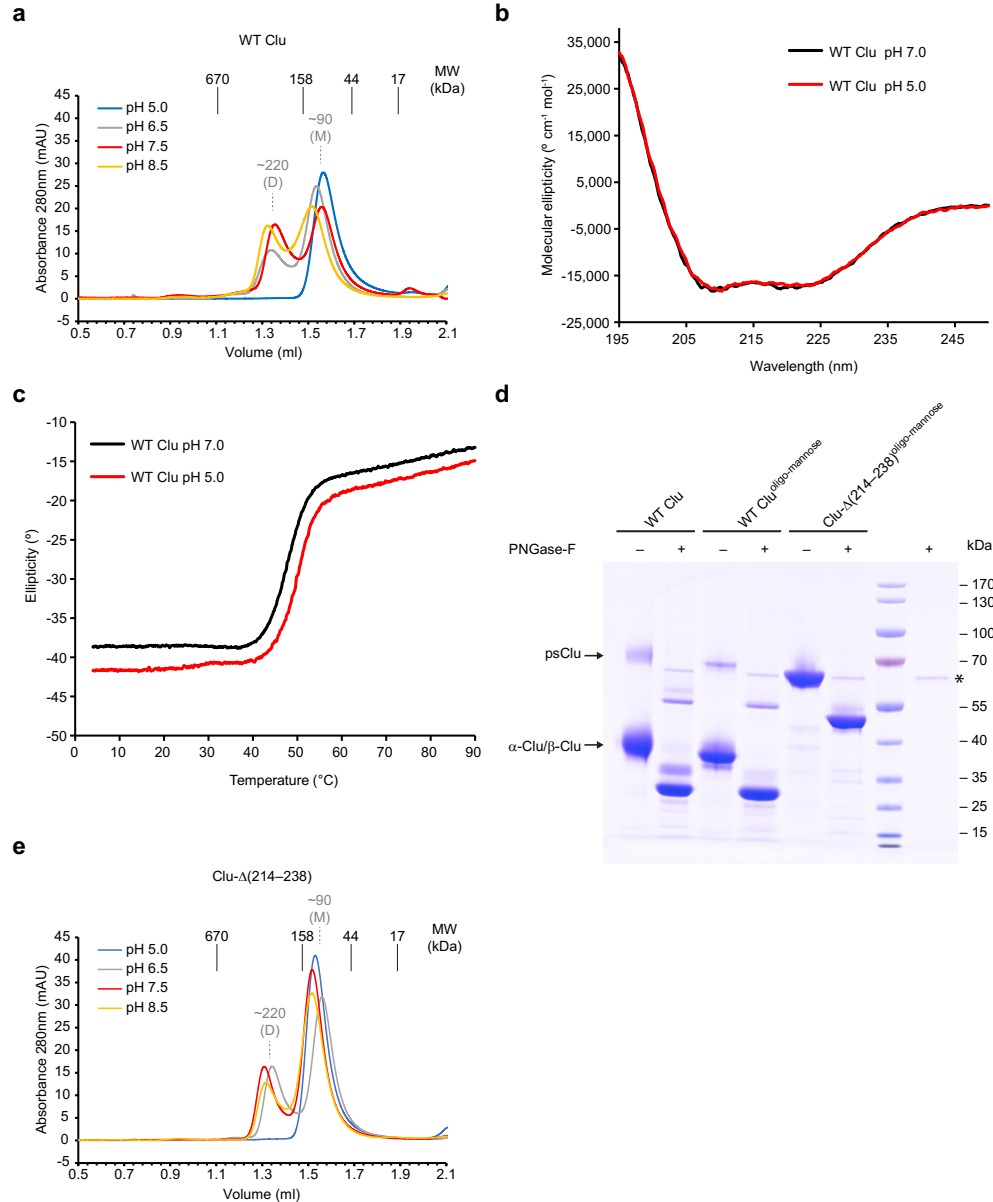

**Extended Data Fig. 1 | Biochemical characterization of recombinant clusterin proteins.** (**a**) pH-dependent oligomerization of WT Clu. WT Clu at 10 µM concentration was incubated on ice with buffer containing 100 mM NaCl and 1 mM EDTA and either 20 mM Na-acetate pH 5.0, 20 mM MES-NaOH pH 6.5, 20 mM HEPES-NaOH pH 7.5 or Tris-HCl pH 8.5 overnight or longer (see Methods for details). The samples were subsequently analyzed by SEC on a Superdex-200 Increase 3.2/300 column at room temperature (RT). Absorbance traces at 280 nm wavelength are shown. The retention volume of molecular weight standards with their mass values in kDa is indicated in black. The apparent molecular weight of Clu is indicated in grey (M: monomers; D: dimers). Note that the apparent size of Clu observed by SEC is larger than expected due to its elongated shape. Representative results are shown (*n* = 3 biological replicates). (**b**) CD spectra of WT Clu at pH 5.0 and pH 7.0. CD spectra were recorded at 0.1 mg ml⁻¹ protein concentration at 20 °C in 50 mM K-phosphate pH 5.0 (red) or pH 7.0 (black). Averages of 3 independently prepared samples are shown. Molecular ellipticities were calculated. (**c**) CD melting curves of WT Clu at pH 5.0 and pH 7.0. The CD signal at 222 nm wavelength was recorded during slow

heating (60 °C h⁻¹) of 5 µM of WT Clu in 50 mM K-phosphate pH 5.0 (red) or pH 7.0 (black). Representative curves are shown (n = 3 experiments). The averages of the melting temperatures are 47.6 and 49.6 °C for pH 7.0 and 5.0, respectively. (**d**) SDS-PAGE analysis of Clu preparations. A Coomassie blue-stained gel is shown. WT Clu with natural N-glycans and the oligo-mannose forms of WT Clu and the Clu-Δ(214–238) mutant produced in presence of the α-mannosidase-I inhibitor kifunensine are shown before (respective left lane) and after (respective right lane) treatment with the glycosidase PNGase-F (GST-PNGase-F), which removes N-glycans and replaces the attachment site Asn with Asp. The bands of the α- and β-chains of WT Clu overlap at ~40 kDa. Residual uncleaved Clu precursor (psClu) is indicated. The WT Clu bands before PNGase-F cleavage are fuzzy because of glycan heterogeneity. On the two rightmost lanes, molecular weight markers and GST-PNGase-F (asterisk) were analyzed. (**e**) pH-dependent oligomerization of the Clu-Δ(214–238) mutant. The experimental setup was identical to that for WT Clu (panel a). Representative results are shown (*n* = 3 biological replicates).

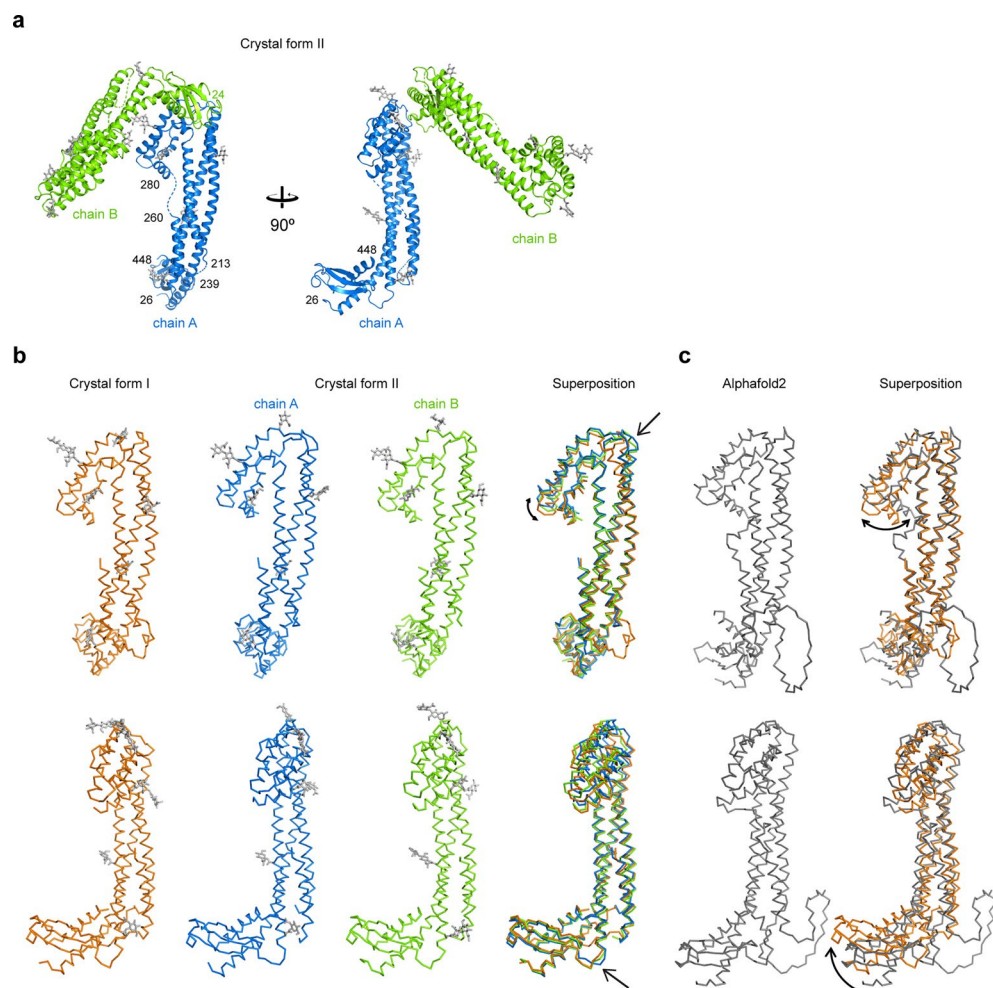

**Extended Data Fig. 2 | Comparison of clusterin crystal structures. (a)** The asymmetric unit of crystal form II. Perpendicular views are shown. Copies A and B of Clu-Δ(214–238) mutant protein are shown as ribbons in blue and green, respectively. First and last ordered residue and gaps in the model are indicated. The ordered parts of N-glycans are shown in stick representation in gray. (**b**) Comparison of crystallographically independent copies of Clu. The three independent copies of Clu-Δ(214–238) mutant protein in crystal forms I and II are shown in orange, blue and green, respectively, on the left. On the right,

a superposition is shown. The peptides are shown as Cα traces and N-glycans as sticks. Arrows indicate regions of local divergence. Two perpendicular views are shown. (**c**) Comparison of the experimental Clu structure with the Alphafold2 model. The Alphafold2 model for human Clu (https://www.alphafold.ebi.ac.uk/entry/P10909) is shown in gray on the left and a superposition with Clu-Δ(214–238) mutant protein in crystal form I on the right. The disulfide domain and the α/β roll-like domain exhibit noticeable reorientations in the Alphafold2 model (indicated by curved arrows).

**a**

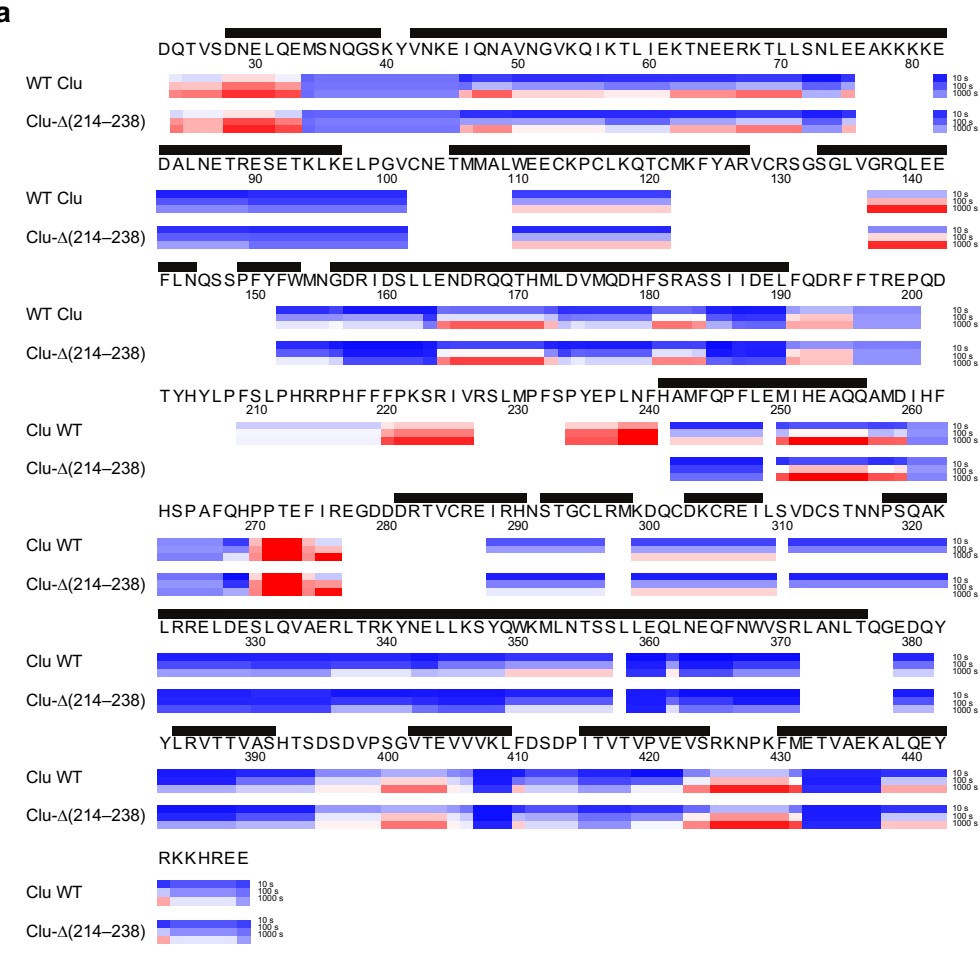

**b**　　　　　　　　　　　　　　　　　　　　**c**

WT Clu　　　　　　　　　　　　　　　　　　Clu-Δ(214–238)

100 s

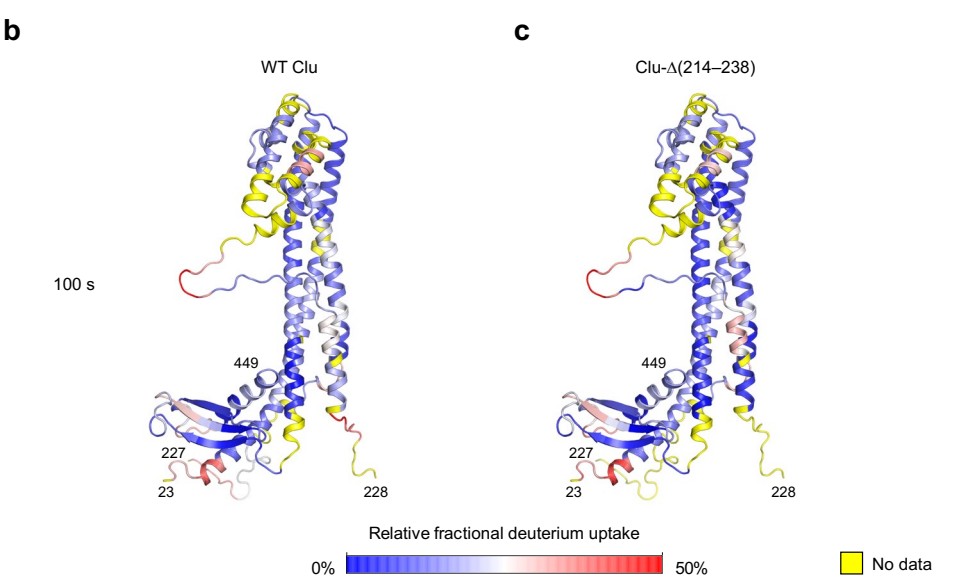

Relative fractional deuterium uptake

0%　　　　　　　　50%　　　　□ No data

**Extended Data Fig. 3 | See next page for caption.**

**Extended Data Fig. 3 | Hydrogen-deuterium exchange analysis of clusterin.**
(**a**) Peptide-level H/DX kinetics for WT Clu (upper rows) and Clu-Δ(214–238) mutant (lower rows) represented as heat maps. WT Clu or Clu-Δ(214–238) mutant at pH 5.0 were exposed to deuterated buffer for 10 s (top), 100 s (middle) and 1000 s (bottom), respectively. After quenching exchange at pH 2.6 and 0 °C, the protein was digested with pepsin and deuterium incorporation into backbone amides of fragment peptides determined by MS. For the analysis, the data from three independent experiments were used. The peptides considered for analysis were identified at least twice. The blue-white-red color gradient represents increasing fractional deuterium incorporation from 0 to 50 %. Secondary structure (black bars), sequence and residue numbers of WT Clu are indicated. Slow and rapid exchange kinetics correlated well with presence of secondary structure and disorder, respectively. For unknown reasons, peptides 196–200, 209–219 and 260–269, predicted to be disordered, exhibited near constant intermediate deuterium levels at all time points. (**b, c**) Mapping of the hydrogen-deuterium exchange data onto the Clu structure model. The fractional deuterium incorporation into peptide segments during 100 s in WT Clu (**b**) and Clu-Δ(214–238) (**c**) was mapped onto the structure model. The same views as in the right panel of Fig. 2c are shown. The blue-white-red color gradient (see scale bar) represents increasing fractional deuterium incorporation from 0 to 50 %. Yellow coloring indicates missing data coverage. N- and C- terminal residues are indicated. Residues 28–33 in helix α1 and residues 137–142 in helix α4 exhibited increased H/D exchange in both samples, consistent with enhanced structural plasticity in solution compared to the crystal lattice.

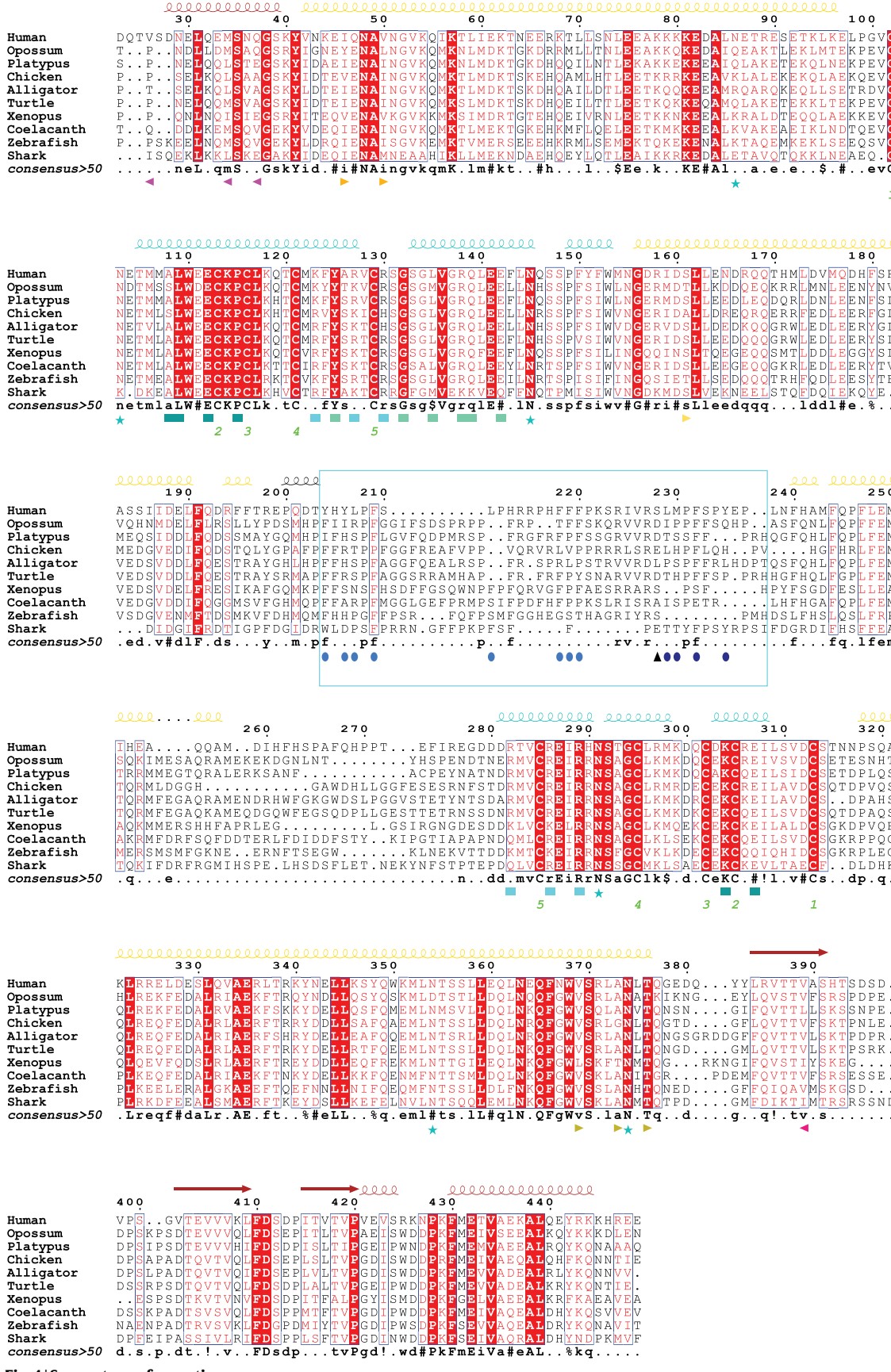

**Extended Data Fig. 4 | See next page for caption.**

**Extended Data Fig. 4 | Alignment of representative clusterin sequences.**
Amino acid sequences of a representative set of Clu homologs were aligned
using the EBI Clustal-Ω server (https://www.ebi.ac.uk/Tools/msa/clustalo/).
Secondary structure elements for human Clu are indicated above the sequences.
The Clu domain structure is indicated by dark red, gold and teal coloring of
secondary structure elements in the α/β roll-like, coiled-coil and disulfide
domain, respectively. Similar residues are shown in red and identical residues
in white on a red background. Blue frames indicate homologous regions. The
consensus sequence is shown at the bottom (Upper case, conserved; lower case,
conserved in more than 50% of the sequences; symbol explanations: #, D/N/E/Q,
$, L/M; %, F/Y;!, I/V). The ER-targeting signal sequences are not shown. Asterisks
in blue below the sequence indicate attachment sites for N-glycans in human
Clu. Disulfide bonds 1–5 are represented by green numbers. Residues mapping
to the disordered tails are indicated by a frame (cyan). The furin cleavage site
(mutated in mutant TL4) is indicated by an arrowhead (black). Hydrophobic
and aromatic residues in the tail regions that were mutated in this study are
indicated by ovals in marine (TL2) and dark blue (TL3). Conserved residues in
the disulfide domain that were substituted in mutants DD1, DD2 and DD3 are
indicated by rectangles in dark teal, turquoise and cyan, respectively. Residues
lining a hydrophobic surface pocket in the α/β roll-like domain are indicated by
backwards arrowheads in magenta (residues mutated in AB1) and hot pink (AB2),
respectively. Forward arrowheads in orange (residues mutated in CC2) and beige
(CC3) indicate a hydrophobic surface patch in the coiled-coil domain. The
golden forward arrowhead indicates Ser residue 161, a residue lining a surface cleft
that was substituted with Arg in mutant CC4. Uniprot accession codes for the
sequences are: P10909, *Homo sapiens* (human); F6UX07, *Monodelphis domestica*
(opossum); F6XFZ6, *Ornithorhynchus anatinus* (platypus); A0A1D5PN61,
*Gallus gallus* (chicken); A0A1U7SNW7, *Alligator sinensis* (chinese alligator);
XP_005284162.1, *Chrysemys picta bellii* (painted turtle); Q6DIX4, *Xenopus
tropicalis* (western clawed frog); H3AKI1, *Latimeria chalumnae* (coelacanth);
Q6PBL3, *Danio rerio* (zebrafish); K4GIQ5, *Callorhinchus milii* (australian
ghostshark). The Figure was prepared with ESPript 3.0[79].

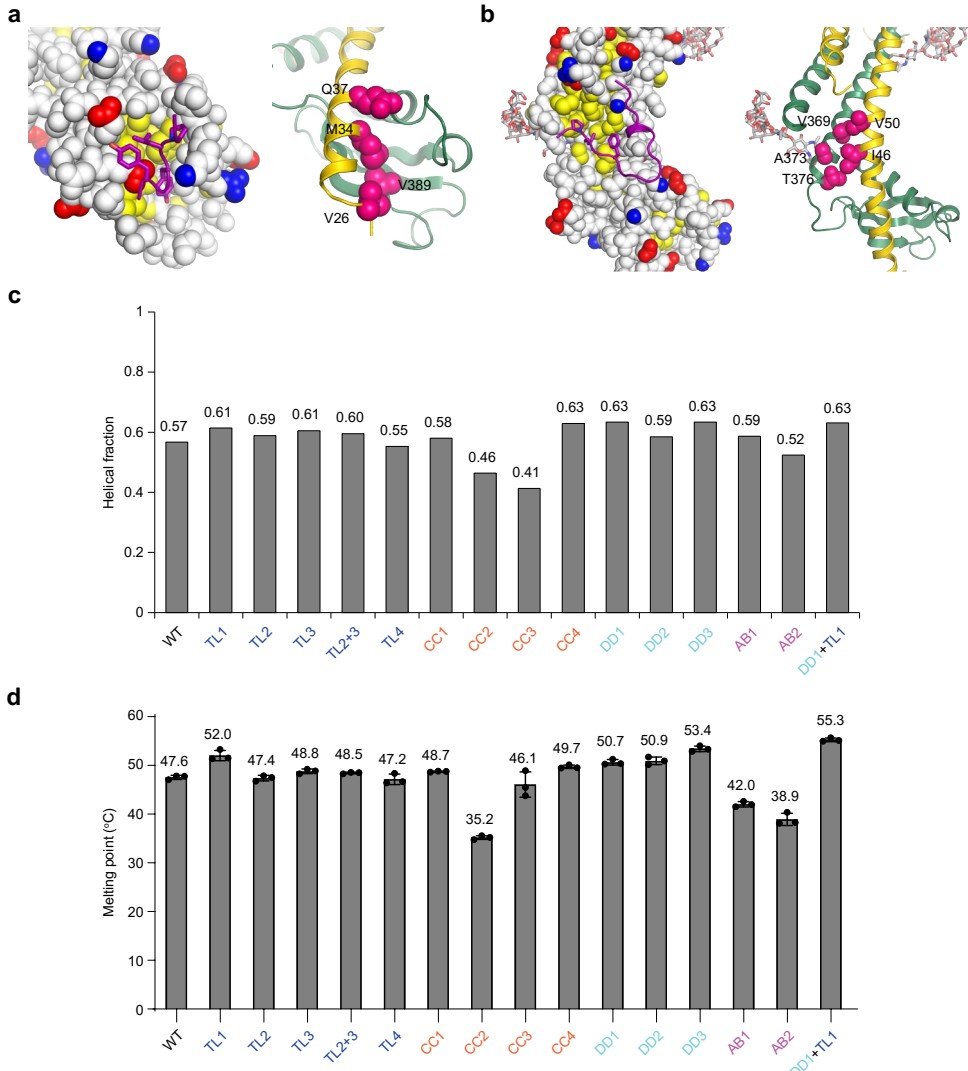

**Extended Data Fig. 5 | Structure-based mutagenesis of human clusterin.**
(**a, b**) Mutations AB1/AB2 and CC2/CC3 of hydrophobic crystal contacts involving the α/β roll-like domain (**a**) and the coiled-coil (**b**). Contact I, mutated in AB1 and AB2 (Fig. 3a), is at the α/β roll-like domain in crystal form I (**a**). Contact II, mutated in CC2 and CC3 (Fig. 3a), is the interface of the loop containing the artificial peptide linkage (between residues 213 and 239, WT Clu numbering) and the coiled coil domain in crystal form I (**b**). In each panel on the left, one Clu chain is shown in space-filling mode with positively and negatively charged functional groups shown in blue and red, respectively, and yellow indicating hydrophobic sidechains. Above the surface, the contacting chain segment is shown as a purple ribbon. Hydrophobic sidechains are shown in stick representation. On the right, a ribbon representation of the model is shown with the substituted residues in mutants AB1 and AB2 (**a**) and CC2 and CC3 (**b**), respectively, highlighted as pink spheres. N-glycans are shown as sticks. (**c**) Bar graph representation of estimated helical fraction of Clu mutant proteins. CD spectra were recorded at a protein concentration of 0.1 mg ml$^{-1}$ at 20 °C in 50 mM K-phosphate pH 7.0. The CD spectra from three independent experiments were averaged, molecular

ellipticities calculated and the helix content (sum of H(r) and H(d)) estimated with the CONTIN algorithm. The deletion mutant TL1 and TL3 as well as most of the mutants with substitutions in or at the disulfide domain (DD1, DD3 and CC4) and the combined mutant DD1 + TL1 exhibited slightly increased helix content. (**d**) Bar graph representation of melting points of Clu mutant proteins. The CD signal at 222 nm wavelength was recorded during slow heating (60 °C h$^{-1}$) of 5 μM of the respective Clu mutant protein in 50 mM K-phosphate pH 7.0. Melting points were estimated using the 'Denatured protein' function of Spectra Manager software (Jasco). Data represents averages ± SD (*n* = 3 biological replicates). Most Clu variants had stability similar to WT Clu, with melting temperatures (T$_m$) of ~47 °C. The tail deletion mutant TL1, the mutants with substitutions in or at the disulfide domain (DD1, DD2, DD3 and CC4) and the combined mutant DD1 + TL1 exhibited slightly increased stability with T$_m$ values of 50.7–55.3 °C, which correlated with increased helix content (see panel c). The hydrophobic pocket mutants AB1, AB2 and CC2 showed reduced stability (T$_m$ values of 35.2–42.0 °C), suggesting that their structure was compromised by the mutations compared to WT Clu.

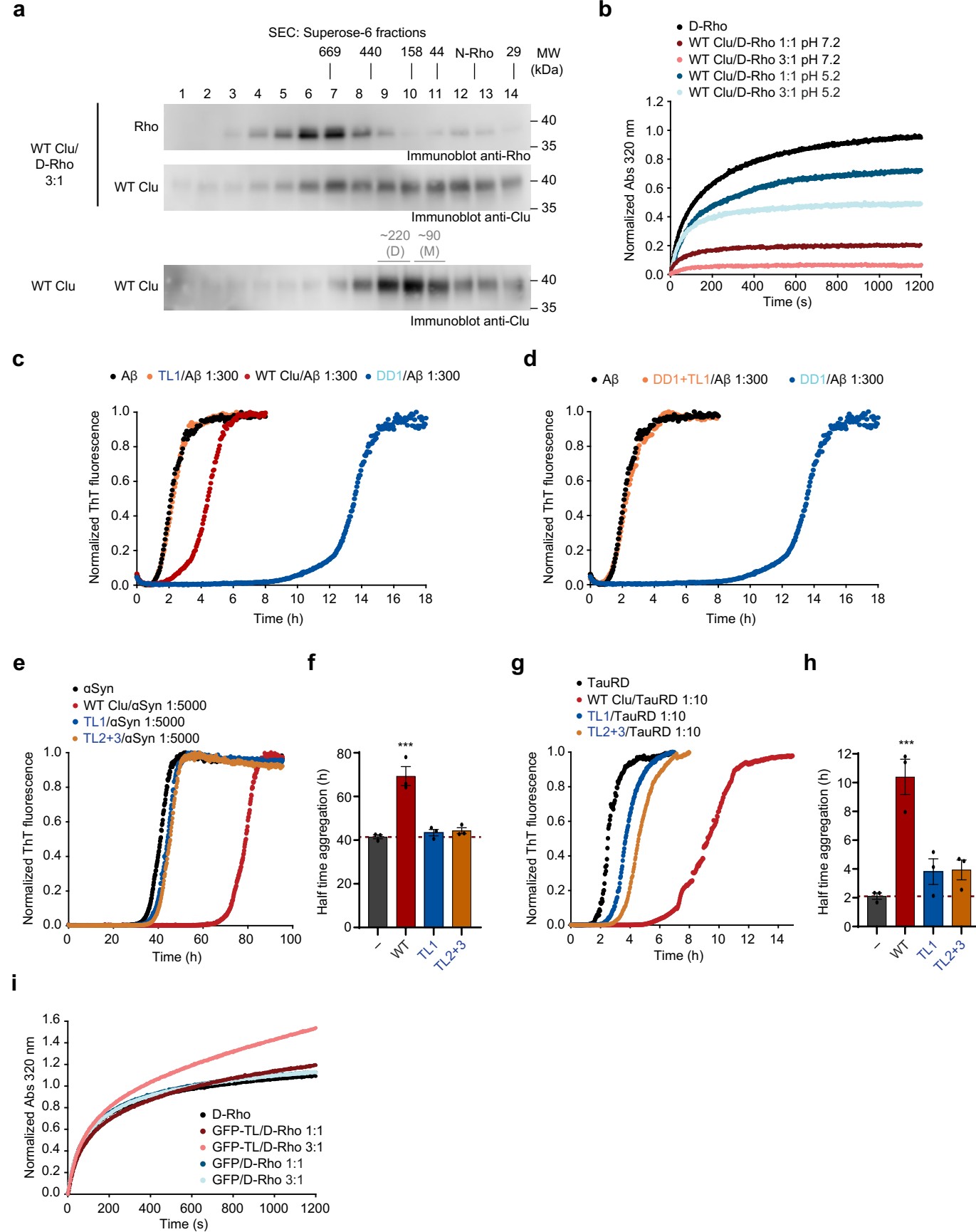

**Extended Data Fig. 6 | See next page for caption.**

**Extended Data Fig. 6 | Clusterin substrate interactions and chaperone activity.**
(**a**) Analysis of Clu–rhodanese complexes by SEC. Denatured rhodanese (D-Rho) was diluted into phosphate-buffered saline (PBS) to a final concentration of 0.5 μM in the presence of WT Clu (1.5 μM), followed by incubation at 25 °C for 30 min. The soluble fraction obtained by centrifugation (15 min, 22,000 g) was analyzed by SEC on Superose-6, followed by SDS-PAGE and immunoblotting against Rho and Clu. A representative immunoblot is shown ($n$ = 3 biological replicates). Of note, no soluble Rho remained after aggregation in absence of Clu. WT Clu alone in PBS is shown for comparison (bottom). The retention volume of marker proteins and their molecular weight in kDa is indicated in black as well as the retention volume of native Rho (N-Rho). The apparent molecular weight of Clu alone is indicated in grey (M: monomers; D: dimers). Note that the apparent size of Clu observed by SEC is larger than expected due to its elongated shape. (**b**) Comparison of Clu holdase activity towards D-Rho at pH 7.2 and 5.2. The assay was performed as described in Fig. 3b, using PBS buffer at pH 7.2 or 20 mM Na-acetate pH 5.2, 150 mM NaCl, 2 mM CaCl$_2$. Representative normalized absorbance traces of D-Rho aggregation in absence of Clu at pH 5.2 (black) or in presence of WT Clu at pH 7.2 (Clu/D-Rho 1:1, dark red; 3:1, light red) and pH 5.2 (Clu/D-Rho 1:1, dark blue; 3:1, light blue) are shown ($n$ = 3 biological replicates). Note that maximum amplitude and kinetics of D-Rho aggregation at pH 7.2 and 5.2 were similar. (**c**) Suppression of Aβ(1–42) amyloid formation by Clu mutant DD1. Aβ amyloid formation was monitored by thioflavin-T (ThT) fluorescence in absence (black) or presence of DD1 mutant (dark blue, molar ratio Clu/Aβ: 1:300) as described in Fig. 3d. Aggregation in presence of WT Clu (red) and mutant

TL1 (orange) at the same molar ratio is shown for comparison. Representative normalized fluorescence traces are shown. ($n$ = 3 biological replicates). (**d**) Effect of Clu mutant DD1 + TL1 on Aβ amyloid formation. Aβ amyloid formation was monitored by ThT fluorescence in absence (black) or presence of DD1 + TL1 mutant (orange, molar ratio Clu/Aβ: 1:300) as described in Fig. 3d. Aβ amyloid formation in presence of mutant DD1 (dark blue) is shown for comparison. (**e-h**) Postponement of α-synuclein and TauRD amyloid formation in presence of Clu variants. Aggregation reactions containing 200 μM α-synuclein (αSyn) (**e**) or 10 μM TauRD (cysteine-free tau, residues 244-372 including two frontotemporal dementia mutations C291A/P301L/C322A/V337M[22]) (**g**) in the absence (black) or presence of WT Clu (red), TL1 mutant (blue) or TL2 + 3 (orange) (molar ratios Clu/αSyn 1:5000 and Clu/TauRD 1:10) were monitored by ThT fluorescence. Representative fluorescence traces are shown. **f, h**) Bar graph showing the relative delay of αSyn (**f**) and TauRD (**h**) aggregation by WT Clu and mutants determined by the half time of reaching the aggregation plateau (red dashed line, αSyn or TauRD alone average). Data represents averages ± s.e.m. ($n$ = 3 biological replicates). *** $P$ < 0.001 by one-way ANOVA with Dunnett's post hoc test comparing Clu/αSyn or Clu/TauRD to αSyn or TauRD alone (WT Clu/αSyn $P$ = 1.146×10$^{-4}$, WT Clu/TauRD $P$ = 3.021×10$^{-4}$). (**i**) Effect of GFP-TL on aggregation of D-Rho. The assay was performed as described in Fig. 3b. Representative normalized absorbance traces of D-Rho alone (black), with additional GFP-TL (1:1, dark red; 3:1, light red) and with GFP as control (1:1, dark blue; 3:1, light blue) are shown ($n$ = 3 biological replicates, except for: D-Rho, n = 4).

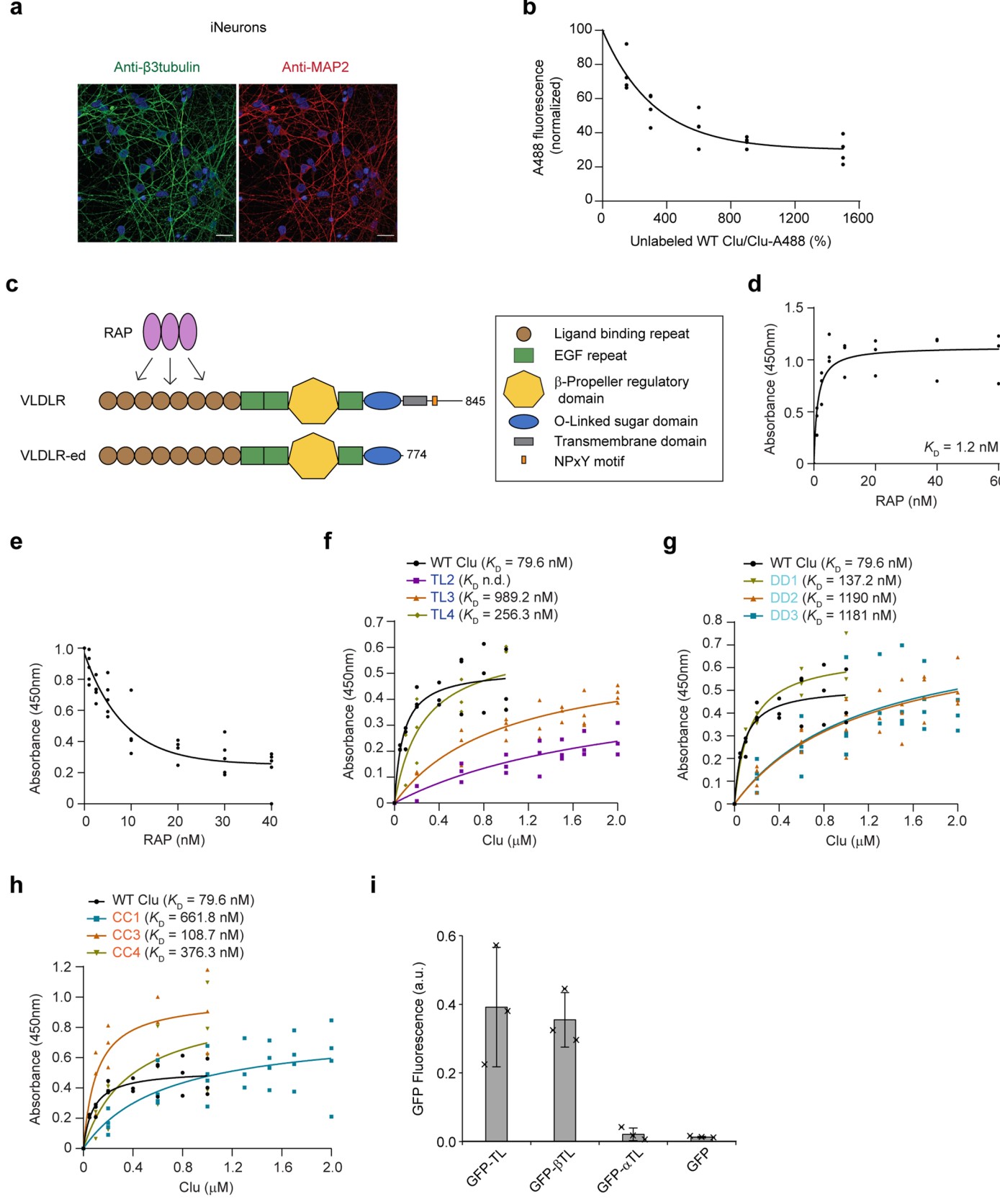

**Extended Data Fig. 7 | See next page for caption.**

**Extended Data Fig. 7 | Cellular uptake and VLDLR receptor binding of WT Clu and mutant proteins.** (**a**) Representative fluorescence microscopy of iNeurons after immunostaining for the neuronal marker proteins β3 tubulin and MAP2. The β3 tubulin and MAP2 signals are green and red, respectively. Nuclear staining with DAPI is shown in blue (n = 2 biological replicates). Scale bars, 20 μm. (**b**) Competition of cellular uptake of Clu-A488 by unlabeled Clu. The fluorescence signal from WT Clu-A488 internalized into iNeurons in the presence of increasing concentrations of unlabeled WT Clu as measured by flow cytometry is shown. Competition curve is shown. Individual data points are plotted (*n* = 4 biological replicates). (**c**) Domain structure of VLDLR and the VLDLR ectodomain (VLDLR-ed) construct. The extracellular LDLR ligand binding repeats, epidermal growth factor (EGF) repeats and β-propeller regulatory domain of VLDLR are shown in brown, green and yellow, respectively. The transmembrane helix is shown in grey. The cytosolic NPxY motif (orange) serves as an internalization signal. The three repeats in low density lipoprotein receptor-related protein-associated protein 1 (RAP, magenta), which acts as a chaperone of LDLR family proteins in the ER, each recognize a pair of LDLR ligand binding repeats[91]. RAP is normally retained in the ER. (**d**) Binding curve of the interaction of RAP with VLDLR-ed as determined by enzyme-linked immunoassay (ELISA) using anti-RAP antibody. ELISA plates were coated with VLDLR-ed, and a series of RAP concentrations was added (1-60 nM). After extensive washing, bound RAP was immunodetected. RAP binding to BSA coated wells was used for a background binding correction. The $K_D$ value and binding curve are shown. Individual data points are plotted (*n* = 3 biological replicates). (**e**) Displacement of Clu from VLDLR-ed by increasing concentrations of RAP determined by ELISA using anti-Clu antibody. ELISA plates were coated with VLDLR-ed and RAP was titrated (1-40 nM) in the presence 100 nM Clu. Clu binding to BSA coated wells was considered background and subtracted. Competition curve is shown. Individual data points are plotted (*n* = 5 biological replicates, except for: 10 nM, *n* = 4). (**f-h**) Affinity of selected Clu mutants to immobilized VLDLR-ed. Binding curves of some of the tail (**f**), DD mutants (**g**) and coiled-coil mutants (**h**) determined by ELISA as described in Fig. 5e. The binding curve of WT Clu is shown for comparison. The $K_D$ values and binding curves are shown (n.d., not determined). Individual data points are plotted (*n* = 3 biological replicates, except for: TL3, DD3, CC1 and CC4, *n* = 4). (**i**) Binding of GFP fusion constructs to VLDLR detected by immune affinity chromatography. The GFP-TL, GFP-βTl and GFP-αTl fusions and VLDLR ectodomain (VLDLR-ed) with a C-terminal C-tag were incubated for 2 h at equimolar concentration in presence of anti-C-tag affinity resin. GFP was used as a reference. Binding in absence of VLDLR-ed served as a background control. After extensive washing, bound proteins were eluted with high-salt buffer and analyzed by GFP fluorescence. Background GFP fluorescence was subtracted. Bars represent averages and error bars standard deviations. (n = 3 biological replicates).

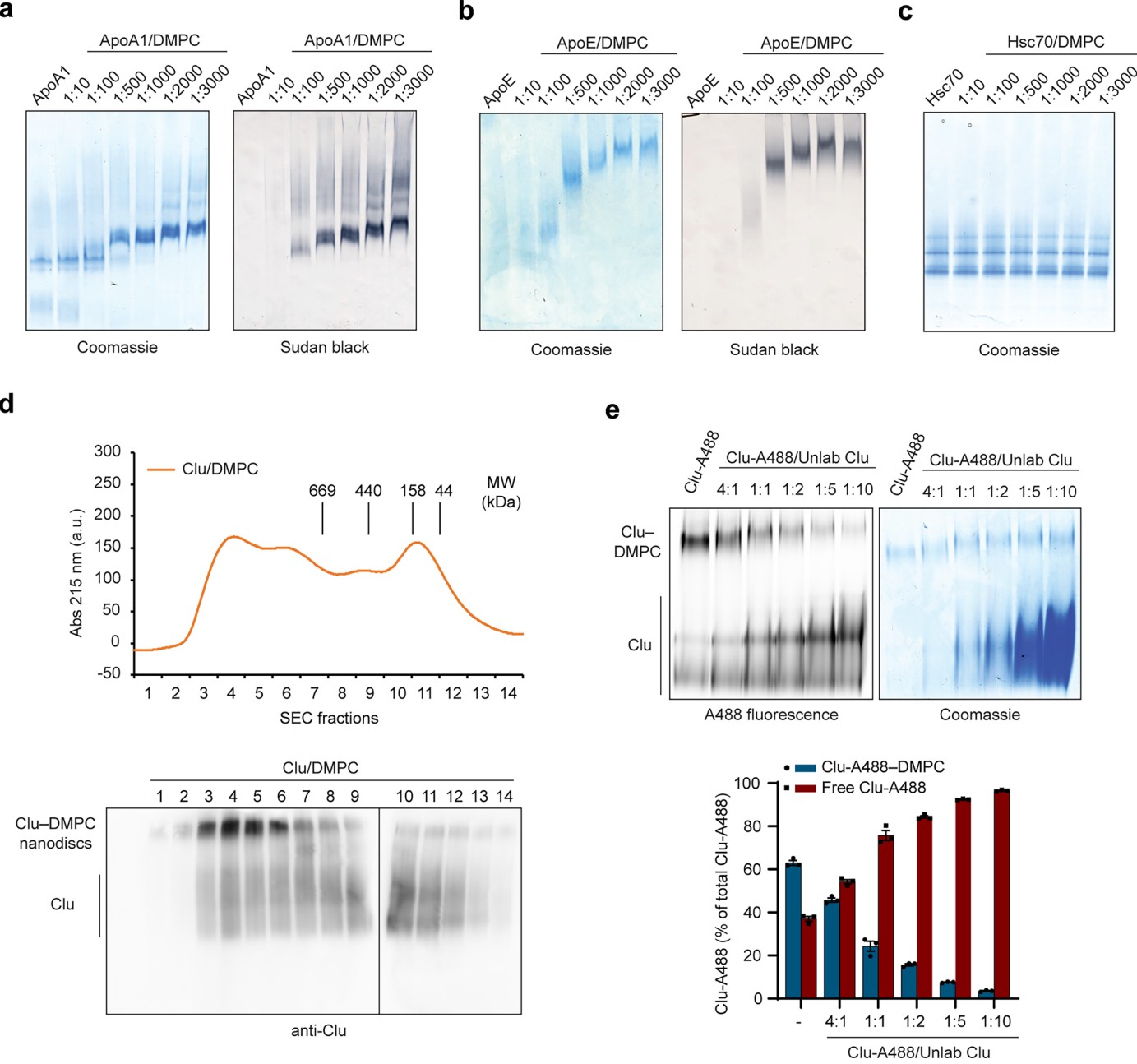

**Extended Data Fig. 8 | Lipoprotein particle formation and dynamics.**
(**a, b**) Formation of lipoprotein particles with ApoA1 (**a**) and ApoE (**b**) detected by native PAGE. Coomassie blue (protein, left) and Sudan black (lipid, right) stained gels are shown. Apolipoprotein and DMPC, at the indicated molar ratios in PBS, were cycled above (30 °C) and below (18 °C) the melt transition temperature of DMPC (24 °C). ApoA1 and ApoE alone are shown for comparison. Representative native gels are shown (*n* = 3 biological replicates). (**c**) Incubation of Hsc70 with the phospholipid DMPC. Human Hsc70 and DMPC at the indicated molar ratios were cycled above (30 °C) and below (18 °C) the melt transition temperature of DMPC (24 °C). Hsc70 alone is shown for comparison. A representative native gel is shown (*n* = 3 biological replicates). (**d**) Fractionation of lipoprotein particles formed from WT Clu and DMPC by SEC. The WT Clu/DMPC reaction mixture was analyzed by analytical SEC on a Superose-6 column equilibrated in PBS. The absorbance trace at 215 nm is shown in the top panel. Fractions were analyzed by native PAGE and immunoblotting. An immunoblot against Clu is shown in the

bottom panel. Clu lipoprotein particles (Clu–DMPC) and free Clu are indicated. A representative experiment is shown (*n* = 3 biological replicates). Free Clu in fractions 3-6 was probably released from lipoprotein particles, as Clu species typically elute later (fractions 9–12). The retention volume of molecular weight markers with their respective mass (in kDa) is indicated. (**e**) Exchange of Clu from lipoprotein particles by free Clu. Purified lipoprotein particles containing Clu-A488 were incubated for 2 h in PBS at 30 °C with increasing concentrations of unlabeled Clu at the indicated molar ratios. The depletion of Clu-A488 from lipoprotein particles was analyzed by native gel and fluorescence detection (top left). A Coomassie-stained native PAGE gel is shown for comparison (top right). Representative gels are shown. The bar graph on the bottom shows the quantification of the fluorescence signals from the Clu–DMPC lipoprotein complex (dark blue) and free Clu (dark red). Data represents averages ± s.e.m. (*n* = 3 biological replicates).

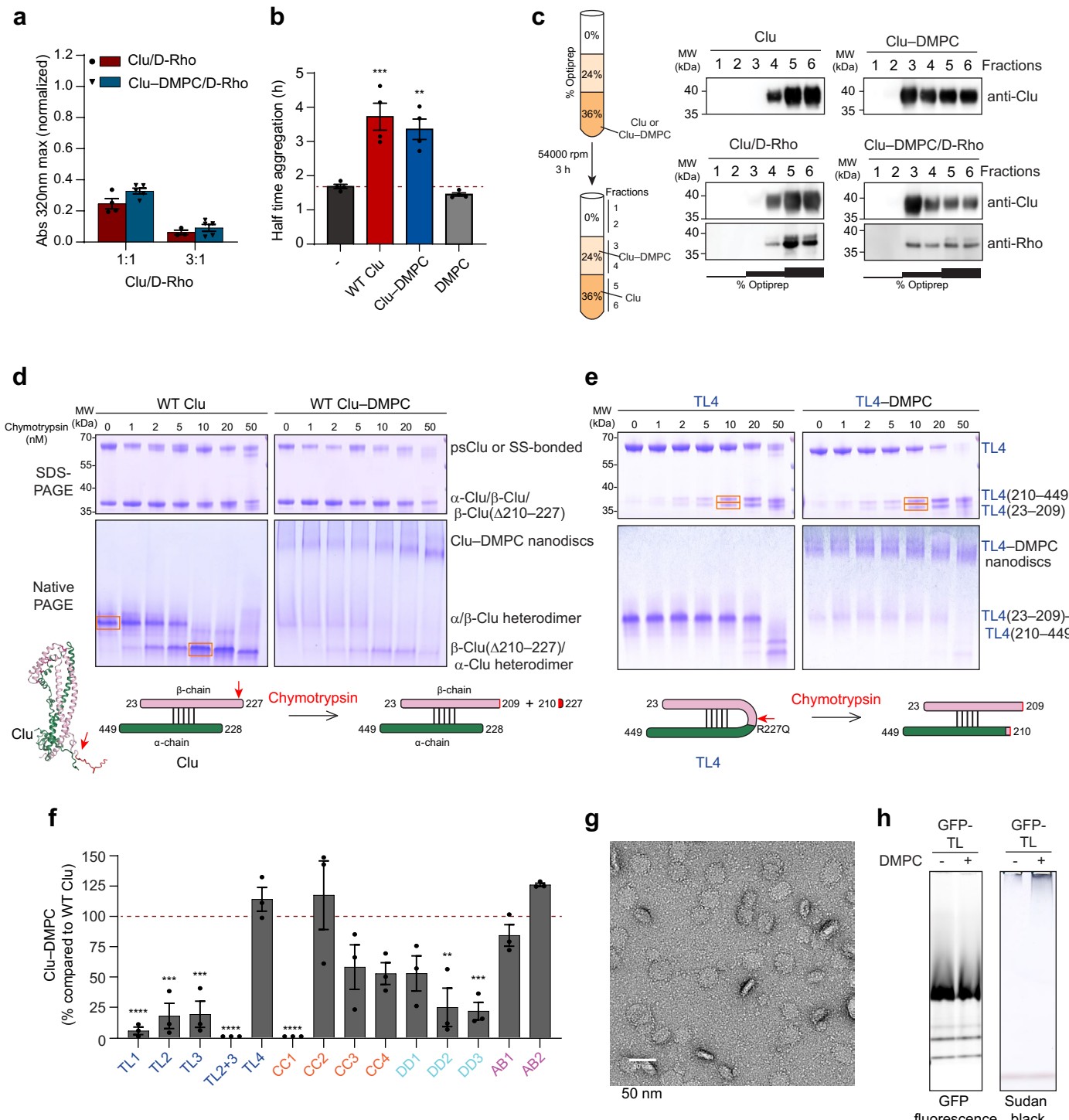

**Extended Data Fig. 9 | See next page for caption.**

**Extended Data Fig. 9 | Functional and structural characterization of Clu lipoprotein particles.** (**a**) Bar graph showing the maximum plateau absorbance of D-Rho aggregation measured by turbidity assay as in Fig. 3c in the presence of Clu or Clu−DMPC. Representative curves are shown in Fig. 6c. Data represents averages ± s.e.m. ($n = 5$ biological replicates, except for: Clu/D-Rho 1:1, $n = 4$, and Clu/D-Rho 3:1, $n = 3$). (**b**) Bar graph showing quantification of Aβ amyloid formation as in Fig. 3e in absence (black) or presence of WT Clu (red), purified Clu lipoprotein complexes (Clu−DMPC, blue) or DMPC (grey) (molar ratio Clu/Aβ: 1:300 or corresponding amount of DMPC). Red dashed line, Aβ aggregation alone. Representative curves are shown in Fig. 6d. DMPC alone had no effect. Data represents averages ± s.e.m. ($n = 4$ biological replicates). ** $P < 0.01$, *** $P < 0.001$ by one-way ANOVA with Dunnett's post hoc test comparing Clu/Aβ to Aβ alone (WT Clu $P = 2.404 \times 10^{-4}$, Clu-DMPC $P = 1.299 \times 10^{-3}$). (**c**) D-Rho binding to Clu−DMPC complexes by flotation assay. D-Rho aggregation reactions were performed with purified Clu lipoprotein complexes or free Clu at a molar ratio of Clu/D-Rho of 3:1 for 30 min in PBS at 25 °C. The reactions were centrifuged and the soluble fraction was mixed with Optiprep to a final concentration of 36%. 1 ml of 24% Optiprep was layered above and subsequently 1 ml of PBS on top. The density gradient stack (left panel) was ultracentrifuged and fractions of 0.5 ml were analyzed by SDS-PAGE and immunoblotting against Clu and Rho. Clu and Clu−DMPC alone were analyzed for comparison. Representative immunoblots are shown on the right ($n = 4$ biological replicates, except for: Clu and Clu−DMPC alone, $n = 3$). Molecular weight (MW) standards are indicated. (**d, e**) Limited proteolysis of free protein versus lipoprotein complexes using WT Clu (**d**) or the single chain mutant TL4 (**e**). Free protein (left) and lipoprotein particles (right) – both of the oligo-mannose form and at 10 μM Clu – were incubated for 30 min in TBS-C buffer at 25 °C with the indicated concentrations of chymotrypsin. After inhibiting proteolysis with 2 mM PMSF on ice, the samples were analyzed by SDS-PAGE (top panels) and native PAGE (bottom panels). Coomassie-stained gels from representative experiments are shown ($n = 3$ biological replicates). Educt and product bands are indicated. (**d**) While proteolytic processing of WT Clu did not become apparent in SDS-PAGE, a large shift was visible in native PAGE for free WT

Clu. LC-MS/MS analysis of the bands indicated by red frames demonstrated that the product consists of the α-Clu/β-Clu-Δ(210-227) heterodimer linked by five disulfide bonds (see scheme below). The most sensitive peptide bond (209-210) is located in the β-tail (see structure representation). The large shift in native PAGE is apparently caused by a change in net charge due to loss of positively charged residues in the β-tail. Of note, the peptide Clu(210-227) (not resolved) bears one Lys and four Arg residues and no Asp/Glu. (**e**) Limited proteolysis of both free TL4 (left) and TL4−DMPC (right) results in two bands near 35 kDa in the SDS-PAGE gels. Proteomics analysis of the bands indicated by red boxes demonstrated that they corresponded to the fragments TL4(23-209) and TL4(210-449), that is the same site is most sensitive to chymotrypsin in both WT Clu and TL4. Because proteolytic nicking (see scheme) does not change the net charge, the electrophoretic mobility of the cleaved product band is identical to that of TL4 in native PAGE. (**f**) Analysis of lipoprotein complex formation by Clu mutants. Formation of lipoprotein complexes with Clu mutant proteins was monitored by native PAGE at 1:500 molar ratio (Clu−DMPC), and the Coomassie blue-stained bands quantified by densitometry (representative gels are shown in Fig. 6e). The bar graph shows the relative intensity of the mutant lipoprotein complex band compared to WT Clu (dashed line). Data represents averages ± s.e.m. ($n = 3$ biological replicates). ** $P < 0.01$, *** $P < 0.001$ and **** $P < 0.0001$ by one-way ANOVA with Dunnett's post hoc test comparing Clu mutants to WT Clu. For exact $P$ values refer to Source Data. (**g**) Analysis of lipoprotein complexes formed with the GFP fusion protein αTL-H7 by negative stain electron microscopy. A representative micrograph of a αTL-H7 lipoprotein complex preparation performed as in Fig. 6f is shown ($n = 3$ biological replicates). The nanodiscs are roughly 25–45 nm in size. The nanodisc rims are decorated with tiny dots, presumably the GFP moieties. Scale bar, 50 nm. (**h**) Native PAGE analysis of GFP-TL interaction with DMPC. Lipoprotein complex formation was performed as in Fig. 6f at a 1:1000 molar ratio (GFP-TL/DMPC) ($n = 2$ biological replicates). GFP fluorescence of the native PAGE gel was analyzed (left) and the gel stained with Sudan black (right). The GFP fluorescence signal of the free protein is saturated.

Andreas Bracher
Patricia Yuste-Checa

# Reporting Summary

## Statistics

For all statistical analyses, confirm that the following items are present in the figure legend, table legend, main text, or Methods section.

| n/a | Confirmed | |
|---|---|---|
| ☐ | ☒ | The exact sample size (*n*) for each experimental group/condition, given as a discrete number and unit of measurement |
| ☐ | ☒ | A statement on whether measurements were taken from distinct samples or whether the same sample was measured repeatedly |
| ☐ | ☒ | The statistical test(s) used AND whether they are one- or two-sided *Only common tests should be described solely by name; describe more complex techniques in the Methods section.* |
| ☒ | ☐ | A description of all covariates tested |
| ☐ | ☒ | A description of any assumptions or corrections, such as tests of normality and adjustment for multiple comparisons |
| ☐ | ☒ | A full description of the statistical parameters including central tendency (e.g. means) or other basic estimates (e.g. regression coefficient) AND variation (e.g. standard deviation) or associated estimates of uncertainty (e.g. confidence intervals) |
| ☐ | ☒ | For null hypothesis testing, the test statistic (e.g. *F*, *t*, *r*) with confidence intervals, effect sizes, degrees of freedom and *P* value noted *Give P values as exact values whenever suitable.* |
| ☒ | ☐ | For Bayesian analysis, information on the choice of priors and Markov chain Monte Carlo settings |
| ☒ | ☐ | For hierarchical and complex designs, identification of the appropriate level for tests and full reporting of outcomes |
| ☒ | ☐ | Estimates of effect sizes (e.g. Cohen's *d*, Pearson's *r*), indicating how they were calculated |

*Our web collection on statistics for biologists contains articles on many of the points above.*

## Software and code

Policy information about availability of computer code

| Data collection | X-ray crystallography diffraction data were collected at beamline ID23-1 of the European Synchrotron Radiation Facility (ESRF), Grenoble, France. The data were integrated and scaled with XDS (Version Feb 5, 2021). The programs Pointless 1.12.4, Aimless 0.7.4 and Ctruncate 1.17.29, as implemented in the CCP4i graphical user interface (Version 7.1.010) were used for data reduction. ThT and ELISA data collection was performed with the SparkControl software v.3.2 (TECAN) ClarioStar software v.5.70 R3 and MARS software V 4.01 R2. Absobance data was recorded using V-500 Control Driver v.1.41.02 (Jasco V-560) Flow cytometry data was collected using the Attune NxT Software v. 5.1.1 Microscopy images were collected using the Leica LAS X software v. 3.5.2 A488 fluorescence signal from gel was collected using the Amersham Typhoon control software 2.0.0.6 Negative staining images were collected using the SerialEM v. 4.1.6 Mass spectrometry proteomics data was recorded on an Orbitrap Exploris 480 with Orbitrap Exploris 480 Tune Application 4.1.355.19 and Xcalibur 4.5 SP1 or on a timsTOF Pro with timsControl 6.0 and Bruker Compass Hystar 6.3 Mass spectrometry lipidomics data was recorded on a QExactive HF with QExactive HF-Orbitrap MS 2.13 build 3162 with Thermo Scientific SII for Xcalibur 1.7.0.468. Immunoblots were develop using an Amersham ImageQuant 800 GxP with the Amersham ImageQuant 800 control software 2.0.0. |
|---|---|
| Data analysis | The crystal structures were solved by molecular replacement with Molrep 11.4.06 (CCP4i interface 7.1.010) using a truncated version of the Alphafold2 model for human clusterin (https://alphafold.ebi.ac.uk/entry/P10909). WinCoot 0.9.4.1 was employed for manual model building. The models were initially refined with Refmac5 5.8.0267 (CCP4i interface 7.1.010). The final refinement was performed with Phenix.refine 1.19.2-4158. Coordinates were aligned with Lsqkab 7.1.010 (CCP4i interface 7.1.010) and Lsqman 081126/9.7.9. Structure drawings and alignments were generated with the programs Pymol 2.2.3 and ESPript 3.0, respectively. |

The alignment of representative Clu sequences was created with the Consurf server (https://consurf.tau.ac.il/consurf_index.php).
Sequences for human alpha-crystallin A and B homologs in jawed vertebrates (taxon ID 7776) were retrieved from Uniprot using BLAST (https://www.uniprot.org/blast) and aligned with Clustalo (https://www.ebi.ac.uk/jdispatcher/msa/clustalo).
The amino acid frequency analysis was performed with Excel 16.0.14332.20788.
ThT and absorbance curve fitting was performed using Sigma plot 14.0 software
Statistical analysis and graphs were generated with GraphPad Prism 10.2.1
Microscopy pictures and gel bands quantification was performed using Image J 1.53a
Peptide search for HDX Mass spectrometry was done using the ProteinLynx Global Server 3.0.2, and subsequently analyzed using the DynamX software 3.0.0
Mass spectrometry proteomics was analyzed using the MaxQuant computational platform v. 2.2.0.0
Mass spectrometry lipidomics was analyzed using the Skyline software v. 23.1.0.455
Data analysis of clusterin uptake was performed using MatLabR2021b (program code at https://github.com/csitron/MATLAB-Programs-for-Flow-Cytometry).
Absorbance coefficients used for protein quantfication were calculated from the protein sequence with the program ProtParam (Swiss Institute of Bioinformatics).

For manuscripts utilizing custom algorithms or software that are central to the research but not yet described in published literature, software must be made available to editors and reviewers. We strongly encourage code deposition in a community repository (e.g. GitHub). See the Nature Portfolio guidelines for submitting code & software for further information.

## Data

Policy information about availability of data

All manuscripts must include a data availability statement. This statement should provide the following information, where applicable:
- Accession codes, unique identifiers, or web links for publicly available datasets
- A description of any restrictions on data availability
- For clinical datasets or third party data, please ensure that the statement adheres to our policy

The datasets, software/code, protocols, and lab materials used and/or generated in this study are listed in a Key Resource Table (Supplementary Table 1) alongside their persistent identifiers at http://doi.org/10.5281/zenodo.14243720. The DOI associated with the diffraction data is 10.15151/ESRF-ES-541098252. The coordinates and structure factors reported in this manuscript have been deposited in the Worldwide Protein Data Bank with accession codes 7ZET and 7ZEU. The mass spectrometry data have been deposited to the ProteomeXchange Consortium via the PRIDE partner repository with the dataset identifiers PXD056940 and PXD057022.

## Research involving human participants, their data, or biological material

Policy information about studies with human participants or human data. See also policy information about sex, gender (identity/presentation), and sexual orientation and race, ethnicity and racism.

| | |
|---|---|
| Reporting on sex and gender | Not applicable |
| Reporting on race, ethnicity, or other socially relevant groupings | Not applicable |
| Population characteristics | Not applicable |
| Recruitment | Not applicable |
| Ethics oversight | Not applicable |

Note that full information on the approval of the study protocol must also be provided in the manuscript.

## Field-specific reporting

Please select the one below that is the best fit for your research. If you are not sure, read the appropriate sections before making your selection.

☒ Life sciences          ☐ Behavioural & social sciences          ☐ Ecological, evolutionary & environmental sciences

For a reference copy of the document with all sections, see nature.com/documents/nr-reporting-summary-flat.pdf

## Life sciences study design

All studies must disclose on these points even when the disclosure is negative.

| | |
|---|---|
| Sample size | No sample size calculation was performed. At least three independent experiments were performed in all cases as per commonly accepted field standards and to enable statistical analysis. No statistical method was used to pre-determine sample sizes. These sample sizes were selected to ensure sufficient data depth and enable robust assessment of differences between conditions. The observed biological effects are clearly detectable and consistently reproduced across replicates, confirming that the sample sizes are sufficient. |

| Data exclusions | Data was excluded just in case technical problems were detected during experiment performance. |
| Replication | A minimum of three independent replicates were conducted for all experiments. Specific number of independent replicates is stated in figure legends. |
| Randomization | Because there is no assignment of data points to distinct groups, randomization did not apply to this study. |
| Blinding | No blinding was performed, as the risk for bias by the experimentalist was deemed irrelevant for this study. |

# Reporting for specific materials, systems and methods

We require information from authors about some types of materials, experimental systems and methods used in many studies. Here, indicate whether each material, system or method listed is relevant to your study. If you are not sure if a list item applies to your research, read the appropriate section before selecting a response.

### Materials & experimental systems

| n/a | Involved in the study |
|---|---|
| ☐ | ☒ Antibodies |
| ☐ | ☒ Eukaryotic cell lines |
| ☒ | ☐ Palaeontology and archaeology |
| ☒ | ☐ Animals and other organisms |
| ☒ | ☐ Clinical data |
| ☒ | ☐ Dual use research of concern |
| ☒ | ☐ Plants |

### Methods

| n/a | Involved in the study |
|---|---|
| ☒ | ☐ ChIP-seq |
| ☐ | ☒ Flow cytometry |
| ☒ | ☐ MRI-based neuroimaging |

## Antibodies

| Antibodies used | anti-MAP2 antibody (AB554, MERCK); anti-β-3-Tubulin (Clone TU-20, MA1-19187, Thermo Fisher Scientific); Goat anti-chicken IgY (H +L) Secondary Antibody Alexa Fluor 647 (A-21449, Thermo Fisher Scientific, 1/500 dilution); F(ab')2-goat anti-mouse IgG (H+L) Cross-Adsorbed Secondary Antibody Alexa Fluor Plus 647 (A48289, Thermo Fisher Scientific, 1/500 dilution); mouse monoclonal Clu-α antibody (Clone B-5, Santa Cruz Biotechnology, sc-5289); rabbit anti-Rhodanese (in-house); CaptureSelect biotin anti-C-tag conjugate (Thermo Fisher Scientific, 7103252100); Conjugated goat-anti mouse immunoglobulin G (IgG)-horseradish peroxidase (HRP) (Merck, A4416); goat-anti rabbit immunoglobulin G (IgG)-horseradish peroxidase (HRP) (Merck, A9169); Streptavidin-HRP (Pierce, 21130) and anti-RAP (Clone E-7, sc-515625 Santa Cruz Biotechnologies) |
| Validation | Most antibodies are validated by commercial suppliers: <br> - anti-MAP2 antibody (AB5543, MERCK): Merck's highly validated antibodies are guaranteed for quality performance. Each batch validated by positive = cerebral cortex/Negative = liver or kidney. Referenced in 41 articles. (https://www.merckmillipore.com/DE/de/product/Anti-MAP2-Antibody,MM_NF-AB5543) <br> - Mouse monoclonal Clu-antibody (Clone B-5, sc-5289, Santa Cruz Biotechnology): Knockdown validation. Referenced in 38 articles. (https://www.scbt.com/p/clusterin-alpha-antibody-b-5) <br> - Anti-β-3-Tubulin (Clone TU-20, MA1-19187, Thermo Fisher Scientific): Knockdown validation. Referenced in 6 articles. (https://www.abcam.com/en-us/products/primary-antibodies/beta-iii-tubulin-antibody-2g10-neuronal-marker-ab78078). <br> - Anti-RAP (Clone E-7, sc-515625 Santa Cruz Biotechnologies): Validated in-house with purified protein. Referenced in 8 articles. (https://www.scbt.com/p/rap-antibody-e-7) <br> - CaptureSelect biotin anti-C-tag conjugate (Thermo Fisher Scientific, 7103252100). EPEA tag technology licensed. (https://www.thermofisher.com/order/catalog/product/7103252100lone%20B-5,%20sc-5289). <br><br> In-house antibody rabbit anti-Rhodanese was validated by binding to purified protein using Western blot. (https://www.antibodyregistry.org/AB_3673130). |

## Eukaryotic cell lines

Policy information about cell lines and Sex and Gender in Research

| Cell line source(s) | HEK293-EBNA suspension cell line was a gift from Yves Durocher (https://doi.org/10.1093/nar/30.2.e9). <br> Induced pluripotent stem cell (iPSC) line HPSI0214i-kucg_2 (RRID:CVCL_AE60, Male) was purchased from UK Health Security Agency (#77650065, supplied by HipSci). |
| Authentication | No further authentication was performed. |
| Mycoplasma contamination | HEK293E were negative for mycoplasma. <br> iPSCs were not tested for Mycoplasma. |
| Commonly misidentified lines (See ICLAC register) | None. |

# Plants

| | |
|---|---|
| Seed stocks | *Report on the source of all seed stocks or other plant material used. If applicable, state the seed stock centre and catalogue number. If plant specimens were collected from the field, describe the collection location, date and sampling procedures.* |
| Novel plant genotypes | *Describe the methods by which all novel plant genotypes were produced. This includes those generated by transgenic approaches, gene editing, chemical/radiation-based mutagenesis and hybridization. For transgenic lines, describe the transformation method, the number of independent lines analyzed and the generation upon which experiments were performed. For gene-edited lines, describe the editor used, the endogenous sequence targeted for editing, the targeting guide RNA sequence (if applicable) and how the editor was applied.* |
| Authentication | *Describe any authentication procedures for each seed stock used or novel genotype generated. Describe any experiments used to assess the effect of a mutation and, where applicable, how potential secondary effects (e.g. second site T-DNA insertions, mosiacism, off-target gene editing) were examined.* |

# Flow Cytometry

## Plots

Confirm that:

☒ The axis labels state the marker and fluorochrome used (e.g. CD4-FITC).

☒ The axis scales are clearly visible. Include numbers along axes only for bottom left plot of group (a 'group' is an analysis of identical markers).

☒ All plots are contour plots with outliers or pseudocolor plots.

☒ A numerical value for number of cells or percentage (with statistics) is provided.

## Methodology

| | |
|---|---|
| Sample preparation | After the corresponding incubation time, cells were placed on ice, washed with PBS and collected with Accutase (Stem Cell technologies). Cells were washed once with PBS, fixed with 4% PFA/PBS for 10 min, washed with PBS, resuspended in 160 µl of PBS and stored at 4 °C until analysis. Right before measuring, 50 µl of Trypan blue solution 0.4% (Thermo Fisher Scientific) were added to each sample to quench the A488 fluorescence outside the cells. |
| Instrument | Attune NxT flow cytometer (Thermo Fisher Scientific). To measure the A488 signal, cells were excited with 488 nm laser light and fluorescence was determined using the 530/30 filter. |
| Software | Attune NxT flow cytometer software v. 5.1.1<br>Data processing was performed using MatLabR2021b |
| Cell population abundance | For each sample at least 10,000 cells were analyzed (average analyzed cells: 47,000). |
| Gating strategy | Cells were gated by size using forward scatter. |

☒ Tick this box to confirm that a figure exemplifying the gating strategy is provided in the Supplementary Information.

