## [Peer Review File · Nature Structural & Molecular Biology]

Structural analyses define the molecular basis of clusterin chaperone function

Corresponding Author: Professor F. Ulrich Hartl

Version 0:

Decision Letter:

28th Jan 2025

Dear Dr. Hartl,

Thank you again for submitting your manuscript "Hydrophobic tails enable diverse functions of the extracellular chaperone clusterin". I apologize for the delay in responding, which resulted from the difficulty in obtaining suitable referee reports. Nevertheless, we now have comments (below) from the 2 reviewers who evaluated your paper. In light of those reports, we remain interested in your study and would like to see your response to the comments of the referees, in the form of a revised manuscript.

You will see that while reviewers appreciate the results, they raise several concerns which will need to be addressed in full in a revision. Specifically, we agree with the reviewers that further experimental support for the internalisation model with strengthen the paper. Furthermore, please explore further mutagenesis in line with suggestions of reviewer #2 to confirm the role of clusterin tails.

Please be sure to address/respond to all concerns of the referees in full in a point-by-point response and highlight all changes in the revised manuscript text file. If you have comments that are intended for editors only, please include those in a separate cover letter.

We are committed to providing a fair and constructive peer-review process. Do not hesitate to contact us if there are specific requests from the reviewers that you believe are technically impossible or unlikely to yield a meaningful outcome, or that you anticipate any issues or delays in preparing the revision within a specified timeline.

We expect to see your revised manuscript within 3 months. If you cannot send it within this time, please contact us to discuss an extension; we would still consider your revision, provided that no similar work has been accepted for publication at NSMB or published elsewhere.

Reporting Summary:

- that unprocessed scans are clearly labelled and match the gels and western blots presented in figures.
- that control panels for gels and western blots are appropriately described as loading on sample processing controls

-- all images in the paper are checked for duplication of panels and for splicing of gel lanes.

EXTENDED DATA FIGURES

Please note that all key data shown in the main figures as cropped gels or blots should be presented in uncropped form, with molecular weight markers. These data can be aggregated into a single supplementary figure item. While these data can be displayed in a relatively informal style, they must refer back to the relevant figures. These data should be submitted with the final revision, as source data, prior to acceptance, but you may want to start putting it together at this point.

Data availability: this journal strongly supports public availability of data. All data used in accepted papers should be available via a public data repository, or alternatively, as Supplementary Information. If data can only be shared on request, please explain why in your Data Availability Statement, and also in the correspondence with your editor. Please note that for some data types, deposition in a public repository is mandatory - more information on our data deposition policies and available repositories can be found below:

<https://www.nature.com/nature-research/editorial-policies/reporting-standards#availability-of-data>

Link Redacted

Sincerely,
Kat

Katarzyna Ciazynska, PhD

(she/her)
Senior Editor
Nature Structural & Molecular Biology
<https://orcid.org/0000-0002-9899-2428>

Reviewers' Comments:

Reviewer #1 (Remarks to the Author):

The manuscript by Yuste-Checa et al. reports extensive analysis of the fascinating extracellular chaperone, clusterin. This chaperone has multiple physiological functions that have been reported for it, and importantly it also inhibits aggregation of a variety of proteins implicated in neurodegenerative diseases. Prior to this study, there was not a structure known for clusterin, so the main result of this paper fills a significant void. Beyond this the authors use mutagenesis to explore structure-function relationships in clusterin, both on its own and in complex with phospholipids. They also describe experiments aimed to understand what is required for clusterin or clusterin-client complexes to be internalized into cells via the very low density lipoprotein receptor. The quality of the data presented is high, and the significance of this work is also high as little is known about this somewhat mysterious chaperone. The main points listed below are intended to point out ways that the manuscript could be made more clear in its presentation. I recommend enough modification to the authors that I call it a major revision. Nonetheless, this is a paper that should be accepted for publication in NSMB due to its impact on the field.

1. It is not clear what the rhodanese aggregation experiments offer. The differences in chaperoning activity of the different mutants for rhodanese versus Abeta or alpha-synuclein raise more questions that do not seem to be relevant to the proposed in vivo action of clusterin. The space that is saved and more focused discussion that would result should help the authors to get their messages across. Also, some of the points raised below will take new main text figures and removing the rhodanese aggregation figures could help free up space. [If the rhodanese aggregation is left in, please put the relevant experimental methods under the same heading as other aggregation methods in the experimental section.]
2. Extended Data Figure 1a should be in the main text with some modifications to show the relationship of the schematic to the domains found by structural study.
3. The mutants studied and codes for them are an alphabet soup that is difficult for a reader to keep track of. Extended Data Figure 5 a and b should be included in the main text with a schematic showing the sequence linearly and the mutations made with codes.
4. The relationship between the chaperone functions of the clusterin tails to their lipid binding needs to be discussed further.
5. The model that a complex between clusterin and a misfolded client is the form that is internalized by the clusterin receptor is attractive. But none of the internalization experimental work was done with this complex but rather clusterin on its own. Why were these experiments not done? They would seem more directly relevant to the proposed function.

Reviewer #2 (Remarks to the Author):

In their manuscript, Yuste-Checa and colleagues analyzed the functional mechanism of the extracellular chaperone clusterin. Based on the crystal structure and mutational analysis, they show a new mechanistic model for the chaperone function of clusterin as well as its cellular uptake. In particular, the authors show that clusterin displays a three-domain architecture with two exposed hydrophobic tails. These tails are necessary for chaperone function. In addition, the tails are engaged in binding to low density lipoprotein receptors and cellular uptake.

Overall, the authors present a highly interesting and very well written manuscript. The experimental data are very sound, of high quality and describe clusterin (structurally and functionally) in an overwhelming, straightforward manner that leads to a new functional model.

Specific comments:

- Fig. 1.: To make it easier for the reader to follow the data presented, it would be helpful to move Extended Data Fig. 1a to main Fig. 1.
- Fig. 2 - Fig.5: The authors propose that both hydrophobic tails are necessary for chaperone function and receptor/lipid binding. To confirm this assumption, data on single-tail mutants (lacking only alpha or beta-tail) are missing. In particular, the alphaTL-GFP-single-tail constructs used in Fig. 4 should be also analyzed for chaperone activity (in Fig. 2) and receptor binding (Fig. 3). According to Fig. 4 and Extended Data Fig. 9, only the beta-tail might be solvent-exposed and sufficient for holdase activity. Therefore, to validate the proposed model (Fig. 5b) it needs to be ruled out that the two tails act separately, with one preferentially binding to substrate proteins and the other interacting with the receptor.
- Fig. 3 and Fig. 5b: The proposed model of the cellular uptake suggests competing affinities of receptor and misfolded protein (substrate) for both tails of clusterin. To verify this competition, the authors might like to analyze the uptake of substrate-preloaded (sub-stoichiometric vs. saturated) clusterin. The authors might comment on the stoichiometry of the substrate-clusterin dimer complexes. According to Extended Data Fig. e 1b and 6b the chaperone activity of monomeric clusterin is diminished. This seems to contradict the proposed model where a clusterin monomer is sufficient for substrate binding. Please explain this in more detail.
- Fig. 4e.: Besides the described effect on clusterin-DMPC-interaction, the authors might like to comment on the broad

distribution of clusterin in the native gels. How would the authors explain the rather distinct banding patterns of some mutants (e.g. TL4, CC1, DD3) in comparison to wt clusterin?

- Fig. 5b: The authors propose a head-to-head orientation for the clusterin monomers in the dimer. This reviewer wonders about the experimental evidence for this assumption and generally misses a structural model of the clusterin dimer. Especially, as the crystal (form II) might indicate a head-to-tail orientation of the monomers in the dimer, the authors need to explain their interpretation of the dimer in more detail.

Reviewer #2 (Remarks on code availability):

The file describing the code used to analyze the results of flow-cytometry is appropriate.

Version 1:

Decision Letter:

Our ref: NSMB-A50106A

24th Apr 2025

Dear Dr. Hartl,

I am writing on behalf of my colleague Dr Ciazynska, as she is out of the office.

Thank you for submitting your revised manuscript "Hydrophobic tails enable diverse functions of the extracellular chaperone clusterin" (NSMB-A50106A). It has now been seen by the original referees and their comments are below. The reviewers find that the paper has improved in revision, and therefore we'll be happy in principle to publish it in Nature Structural & Molecular Biology, pending minor revisions to comply with our editorial and formatting guidelines.

We are now performing detailed checks on your paper and will send you a checklist detailing our editorial and formatting requirements in about 1-2 weeks. Please do not upload the final materials and make any revisions until you receive this additional information from us.

To facilitate our work at this stage, it is important that we have a copy of the main text as a word file. If you could please send along a word version of this file as soon as possible, we would greatly appreciate it; please make sure to copy the NSMB account (cc'ed above).

Thank you again for your interest in Nature Structural & Molecular Biology. Please do not hesitate to contact me if you have any questions.

Sincerely,

Melina Casadio, PhD
Locum Chief Editor, Nature Structural & Molecular Biology
ORCID ID: <https://orcid.org/0000-0003-2389-2243>

Reviewer #1 (Remarks to the Author):

The revised manuscript addressed all points raised in my earlier review satisfactorily. The manuscript is now acceptable.

Reviewer #2 (Remarks to the Author):

The authors answered all questions raised. Incorporating some new data and changing some figures and text improved the revised manuscript which now seems acceptable for publication.

Response to reviewers

We thank the reviewers for their thoughtful comments that helped us greatly in improving the manuscript.

Reviewer #1 (Remarks to the Author):

The manuscript by Yuste-Checa et al. reports extensive analysis of the fascinating extracellular chaperone, clusterin. This chaperone has multiple physiological functions that have been reported for it, and importantly it also inhibits aggregation of a variety of proteins implicated in neurodegenerative diseases. Prior to this study, there was not a structure known for clusterin, so the main result of this paper fills a significant void. Beyond this the authors use mutagenesis to explore structure-function relationships in clusterin, both on its own and in complex with phospholipids. They also describe experiments aimed to understand what is required for clusterin or clusterin-client complexes to be internalized into cells via the very low density lipoprotein receptor. The quality of the data presented is high, and the significance of this work is also high as little is known about this somewhat mysterious chaperone. The main points listed below are intended to point out ways that the manuscript could be made more clear in its presentation. I recommend enough modification to the authors that I call it a major revision. Nonetheless, this is a paper that should be accepted for publication in NSMB due to its impact on the field.

1. It is not clear what the rhodanese aggregation experiments offer. The differences in chaperoning activity of the different mutants for rhodanese versus A β or alpha-synuclein raise more questions that do not seem to be relevant to the proposed *in vivo* action of clusterin. The space that is saved and more focused discussion that would result should help the authors to get their messages across. Also, some of the points raised below will take new main text figures and removing the rhodanese aggregation figures could help free up space. [If the rhodanese aggregation is left in, please put the relevant experimental methods under the same heading as other aggregation methods in the experimental section.]

Clusterin is thought to prevent the deposition and help in the clearance of both fibrillar and non-fibrillar aggregates. Rhodanese is a suitable model substrate for the latter category. In the assays with Amyloid-beta we specifically analyze inhibition of nucleation-dependent amyloid structure formation, not prevention of aggregation in general. The difference between both reactions is that with Amyloid-beta, Clu is active at substoichiometric concentration in delaying nucleation-dependent fibril formation, whereas rhodanese aggregation prevention requires equimolar and superstoichiometric Clu concentrations for holdase function. We therefore consider the rhodanese data relevant. This is now clarified in the revised manuscript (lines 143-144; 167; 343-344).

We thank the reviewer for the suggestion and now we include all aggregation assays under the same experimental section "Protein aggregation reactions and thioflavin-T (ThT) fluorescence measurement".

2. Extended Data Figure 1a should be in the main text with some modifications to show the relationship of the schematic to the domains found by structural study.

We thank the reviewer for suggesting this (also pointed out by reviewer #2). We have moved this panel into the main text as new Fig. 1. To indicate the location of the discontinuous domain segments in the Clu chains, we have added a schematic as new Fig. 2b. Adding the domain structure to Fig. 1 as well would have made the figure very information dense. A linear representation of the domain structure together with the sequence is also shown in the alignment in Extended Data Fig. 4 (Domain coloring of secondary structure elements).

3. The mutants studied and codes for them are an alphabet soup that is difficult for a reader to keep track of. Extended Data Figure 5 a and b should be included in the main text with a schematic showing the sequence linearly and the mutations made with codes.

We believe that naming the mutants according to the affected domain accompanied by a 3D-representation (right panel of what is now Fig. 3a) is most straightforward, because residues contributing to a continuous site might be far apart in sequence owing to the convoluted fold topology of clusterin (for example the adjacent E112K and K304A in mutant DD1). For better overview we assigned the mutation sites in the clusterin sequence alignment in Extended Data Fig. 4. Additionally, we have included a new column in the clusterin mutant table (New Fig. 3a) indicating the intended target of the mutations, so the reader can easily keep track of the reasoning behind the mutants.

We have shown the strategy behind mutations AB1, AB2, CC2 and CC3 only in the supplement, because our speculations about accessory substrate binding sites of clusterin in these regions did not pan out. The contacts in the crystal lattice we referred to seem to mainly result from the high clusterin concentration in the crystallization experiment. This is now clarified in lines 162-164.

4. The relationship between the chaperone functions of the clusterin tails to their lipid binding needs to be discussed further.

This is a valid point since the α -tail may be engaged in stabilizing the nanodiscs together with helix $\alpha 7$ and thus may not be available for client protein interactions. Our new data with GFP fusion proteins (experiments proposed by reviewer #2) (new Fig. 3f-h) show that the β -tail alone has only weak chaperone activity compared to both tails attached to the same GFP, indicating that simultaneous action of both tails is required for efficient chaperone function. We suggest that the chaperone activity of the clusterin-lipoprotein complexes can be explained by clusterin forming dimeric units on the nanodisc which either expose both the α - and β -tails of one subunit or the β -tails from adjacent subunits for cooperative substrate interactions. The latter possibility might be relevant if the α -tail structurally cooperates with helix $\alpha 7$ for clusterin-lipid binding, in which case the α -tail may not be exposed. This is now explained in the Conclusion section of the revised manuscript in lines 344-358. For completeness, we now also include a lipoprotein complex formation assay in presence of GFP-TL (New Extended Data Fig. 9h). This shows that the tails are not sufficient for nanodisc formation, consistent with the additional requirement of helix $\alpha 7$.

5. The model that a complex between clusterin and a misfolded client is the form that is internalized by the clusterin receptor is attractive. But none of the internalization experimental work was done with this complex but rather clusterin on its own. Why were these experiments not done? They would seem more directly relevant to the proposed function.

Thank you for this suggestion, which was also made by reviewer #2. We have now included the requested uptake experiments for WT clusterin and mutants in the presence of A β aggregates as new Fig. 4a and b (lines 207-226 in the text). We found that the effects of clusterin tail mutants on uptake show a generally similar trend in the absence or presence of substrate, consistent with their critical role in uptake. Interestingly, WT and mutant clusterin uptake is stimulated and not diminished in the presence of substrate, contrary to what might have been expected considering that the hydrophobic tails are necessary for both clusterin uptake and chaperone activity (Fig. 3b-h and new Fig. 4a, b). Similar results showing stimulated cell uptake of WT clusterin have previously been reported with denatured luciferase as substrate (Itakura et al. 2020). We interpret this behavior in support of the model that oligomeric clusterin-client protein complexes are the substrate for uptake, where some of the hydrophobic clusterin tails are engaging uptake receptors while others are available for client protein binding (New Fig. 6b). This could plausibly occur if clusterin formed dimeric units in such oligomers (as also mentioned in comment #4 in the context of lipid binding).

Reviewer #2 (Remarks to the Author):

In their manuscript, Yuste-Checa and colleagues analyzed the functional mechanism of the extracellular chaperone clusterin. Based on the crystal structure and mutational analysis, they show a new mechanistic model for the chaperone function of clusterin as well as its cellular uptake. In particular, the authors show that clusterin displays a three-domain architecture with two exposed hydrophobic tails. These tails are necessary for chaperone function. In addition, the tails are engaged in binding to low density lipoprotein receptors and cellular uptake.

Overall, the authors present a highly interesting and very well written manuscript. The experimental data are very sound, of high quality and describe clusterin (structurally and functionally) in an overwhelming, straightforward manner that leads to a new functional model.

Specific comments:

- Fig. 1.: To make it easier for the reader to follow the data presented, it would be helpful to move Extended Data Fig. 1a to main Fig. 1.

We thank the reviewer for this suggestion also pointed out by reviewer #1. We have moved Extended Data Fig. 1a into the main text as new Fig. 1.

- Fig. 2 - Fig.5: The authors propose that both hydrophobic tails are necessary for chaperone function and receptor/lipid binding. To confirm this assumption, data on single-tail mutants (lacking only alpha or beta-tail) are missing. In particular, the alphaTL-GFP-single-tail constructs used in Fig. 4 should be also analyzed for chaperone activity (in Fig. 2) and receptor binding (Fig. 3). According to Fig. 4 and Extended Data Fig. 9, only the beta-tail might be solvent-exposed and sufficient for holdase activity. Therefore, to validate the proposed model (Fig. 5b) it needs to be ruled out that the two tails act separately, with one preferentially binding to substrate proteins and the other interacting with the receptor.

We think that the mutants TL2 (substitution of hydrophobics with Ser in β -tail) and TL3 (substitution of hydrophobics with Ser in α -tail) serve as a proxy for the activity of the isolated tails in the context of the clusterin backbone, as shown by the similar results of TL1 (deletion of both tails) and TL2+3 (substitution of hydrophobics with Ser in both α - and β -tails) mutants in all assays. The activities of TL2 and TL3 are consistent with our model that both tails contribute to chaperone function, receptor binding and lipoprotein complex formation and no clear differential preference of the α - or the β -tail for a specific function is observed. The β -tail is twice as long as the α -tail (containing twice as many hydrophobic residues) and exhibits a stronger contribution in all assays. In light of the data already available we feel that the additional information, if any, gained from analysis of clusterin mutants with single tail deletions, would not justify the major effort necessary to express and purify these mutant proteins.

However, to test the function of the α - and β -tails in isolation, we have constructed and expressed a new GFP fusion protein with only the β -tail (GFP- β TL = GFP-Clu(204–227)) and then analyzed GFP- β TL, GFP- α TL and GFP-TL for A β amyloid inhibition, cellular uptake and binding to VLDLR-ed. These new data are now included in the manuscript (new Fig. 3f-h, new Fig. 4c and new Extended Data Fig. 7i, lines 188-198, 227-232 and 254-257). Of note, we were unable to estimate the VLDLR-ed binding affinities by the ELISA assay used for clusterin due to high background signal from unspecific binding of GFP-TL to the BSA-coated plate, and had to use a modified binding assay. Our results show that the combined tails in GFP-TL are sufficient to confer receptor binding, cell uptake and chaperone activity, consistent with our model. GFP- β TL had residual chaperone and uptake activity, while exhibiting binding to VLDLR-ed similar to GFP-TL. GFP- α TL was as inactive as GFP in all assays, but combined with the amphiphilic helix α 7 (GFP- α TL-H7) was sufficient for lipoprotein complex formation. Thus, the data are consistent with a requirement for the tails functioning synergistically in chaperone activity and uptake into iNeurons. For VLDLR binding the β -tail appears to be more important. We suggest the involvement of clusterin dimers (or higher oligomers) in such ternary complexes, where one clusterin protomer may interact with substrate and the other with receptor or lipid. Clusterin dimers form readily at neutral pH independently of the tails (Extended Data Fig. 1a, e). This is now discussed in the Conclusion section of the revised manuscript in lines 344-358.

- Fig. 3 and Fig. 5b: The proposed model of the cellular uptake suggests competing affinities of receptor and misfolded protein (substrate) for both tails of clusterin. To verify this competition, the authors might like to analyze the uptake of substrate-preloaded (sub-stoichiometric vs. saturated) clusterin.

We have now included the uptake experiment of WT and clusterin mutants in the presence of A β aggregates (saturating conditions, new Fig. 4a and b and lines 207-226). We thank both reviewers for this suggestion (also see response to reviewer #1 point 5).

Interestingly, we observed a stimulation of clusterin uptake into iNeurons by A β aggregates in a concentration-dependent manner (at concentration ratios of clusterin to A β , 1:3 to 1:10). These results align with our model that clusterin may function as a dimer or higher oligomer in client protein binding and cellular uptake. We hypothesize that clusterin uptake is stimulated when multiple clusterin molecules bind to oligomeric client protein. Additionally, passive uptake of A β -bound clusterin via specific A β aggregate receptors may also occur, nor can the possibility be ruled out that substrate binding induces a structural rearrangement in clusterin, enhancing its receptor interaction. These findings highlight the complexity of this process and underscore the need for further studies, beyond the scope of this manuscript, to elucidate the exact uptake mechanism.

The authors might comment on the stoichiometry of the substrate-clusterin dimer complexes. According to Extended Data Figs. 1b and 6b the chaperone activity of monomeric clusterin is diminished. This seems to contradict the proposed model where a clusterin monomer is sufficient for substrate binding. Please explain this in more detail.

The actual clusterin-substrate complexes are likely small substrate aggregates interacting with multiple clusterin molecules. In the SEC analysis shown in Extended Data Fig. 6a, clusterin (Clu) (MW ~75 kDa) is apparently in excess to denatured rhodanese (D-Rho) (~37 kDa) in the ~700 kDa Clu:D-Rho complexes, given the 3:1 Clu:D-Rho input ratio. Consistently, we previously showed an interaction of clusterin with tau oligomers but not tau monomer (Yuste-Checa et al., *Nat Commun*, 2021). While monomeric clusterin is probably capable of interacting with substrate in a dynamic complex, dimeric (or oligomeric) clusterin seems to be more efficient, due to higher avidity. In such complexes, clusterin dimers might be stabilized by the high local clusterin concentration on the aggregate.

For binding of such assemblies to receptors, the clusterin dimers would have to reorganize to liberate some tails. To reflect this better, our revised model in new Fig. 6b shows the simultaneous interaction of multiple clusterin dimers both with substrate and clusterin receptors. An additional putative direct interaction of substrate with an alternate receptor is also shown (as discussed in the previous comment).

- Fig. 4e.: Besides the described effect on clusterin-DMPC-interaction, the authors might like to comment on the broad distribution of clusterin in the native gels. How would the authors explain the rather distinct banding patterns of some mutants (e.g. TL4, CC1, DD3) in comparison to wt clusterin?

We believe that the broad distribution of clusterin in native PAGE is caused by glycan heterogeneity. Of note, the oligo mannose forms of WT-clusterin and TL4 mutant have a sharper band appearance (Extended Data Fig. 9d, e). The downshift of TL1 and DD3 relative to WT-clusterin is consistent with increased electrophoretic mobility upon loss of net positive charge (-5 for TL1, -7 for DD3 compared to WT). The TL4 and CC1 band patterns appeared peculiar only in this repeat of the experiment. We are now showing a more representative result in new Fig. 5e of the revised manuscript (see replicate gels in Source Data file).

- Fig. 5b: The authors propose a head-to-head orientation for the clusterin monomers in the dimer. This reviewer wonders about the experimental evidence for this assumption and generally misses a structural model of the clusterin dimer. Especially, as the crystal (form II) might indicate a head-to-tail orientation of the monomers in the dimer, the authors need to explain their interpretation of the dimer in more detail.

The lattice of crystal form II has both asymmetric interfaces – as in the asymmetric unit shown in Extended Data Fig. 2a – and symmetrical (along a two-fold crystallographic axis) interfaces (data not shown). The latter symmetrical clusterin dimers have head-to-head parallel architectures with interactions between the coiled-coils, but analysis with Pisa indicated that their contact interfaces (buried surface areas of 471 and 402 Å²) are too small for stable complex formation. The same was the case for the asymmetric interfaces.

Of note, Alphafold3 models show head-to-head architectures as well (with interactions between the disulfide bond domain and the coiled coil), but do look rather implausible and have poor pTM and ipTM scores (data not shown). It is also possible that clusterin undergoes some structural rearrangement upon dimer formation. This remains to be explored.

We arbitrarily proposed a head-to-head, parallel structure because this would bring the tails from adjacent protomers into proximity for functional cooperation in client protein and receptor binding. This is now clarified in lines 356-358. We are working towards a high-resolution structure of the clusterin dimer – however so far without success.

Reviewer #2 (Remarks on code availability):

The file describing the code used to analyze the results of flow-cytometry is appropriate.
This email has been sent through the Springer Nature Tracking System NY-610A-NPG&MTS